# BrainPy, a flexible, integrative, efficient, and extensible framework for general-purpose brain dynamics programming

Chaoming Wang[1,2], Tianqiu Zhang[1], Xiaoyu Chen[1], Sichao He[3], Shangyang Li[1], Si Wu[1,2]*

[1]School of Psychological and Cognitive Sciences, IDG/McGovern Institute for Brain Research, Peking-Tsinghua Center for Life Sciences, Center of Quantitative Biology, Academy for Advanced Interdisciplinary Studies, Bejing Key Laboratory of Behavior and Mental Health, Peking University, Beijing, China; [2]Guangdong Institute of Intelligence Science and Technology, Guangdong, China; [3]Beijing Jiaotong University, Beijing, China

**Abstract** Elucidating the intricate neural mechanisms underlying brain functions requires integrative brain dynamics modeling. To facilitate this process, it is crucial to develop a general-purpose programming framework that allows users to freely define neural models across multiple scales, efficiently simulate, train, and analyze model dynamics, and conveniently incorporate new modeling approaches. In response to this need, we present BrainPy. BrainPy leverages the advanced just-in-time (JIT) compilation capabilities of JAX and XLA to provide a powerful infrastructure tailored for brain dynamics programming. It offers an integrated platform for building, simulating, training, and analyzing brain dynamics models. Models defined in BrainPy can be JIT compiled into binary instructions for various devices, including Central Processing Unit, Graphics Processing Unit, and Tensor Processing Unit, which ensures high-running performance comparable to native C or CUDA. Additionally, BrainPy features an extensible architecture that allows for easy expansion of new infrastructure, utilities, and machine-learning approaches. This flexibility enables researchers to incorporate cutting-edge techniques and adapt the framework to their specific needs.

*For correspondence:
siwu@pku.edu.cn

**Competing interest:** The authors declare that no competing interests exist.

## Editor's evaluation

The paper introduces a new, important framework for neural modelling that promises to offer efficient simulation and analysis tools for a wide range of biologically-realistic neural networks. It provides convincing support for the ease of use, flexibility, and performance of the framework, and features a solid comparison to existing solutions in terms of accuracy. The work is of potential interest to a wide range of computational neuroscientists and researchers working on biologically inspired machine learning applications.

## Introduction

Brain dynamics modeling, which uses computational models to simulate and elucidate brain functions, is receiving increasing attention from researchers across different disciplines. Recently, gigantic projects in brain science have been initiated worldwide, including the BRAIN Initiative (*Jorgenson et al., 2015*), Human Brain Project (*Amunts et al., 2016*), and China Brain Project (*Poo et al., 2016*), which are continuously producing new data about the structures and activity patterns of neural systems. Computational modeling is a fundamental and indispensable tool for interpreting this vast amount of

data. However, to date, we still lack a general-purpose programming framework for brain dynamics modeling. By general purpose, we mean that such a programming framework can implement most brain dynamics models, integrate diverse modeling demands (e.g., simulation, training, and analysis), and accommodate new modeling approaches constantly emerging in the field while maintaining high-running performance. General-purpose programming frameworks are exemplified by Tensor-Flow (*Abadi et al., 2016*) and PyTorch (*Paszke et al., 2019*) in the field of Deep Learning, which provides convenient interfaces for researchers to define various AI models flexibly and efficiently. These frameworks have become essential infrastructure in AI research, and play an indispensable role in this round of the AI revolution (*Dean, 2022*). Brain dynamics modeling also needs such a general-purpose programming framework urgently (*D'Angelo and Jirsa, 2022*).

To develop a general-purpose programming framework for brain dynamics modeling, we face several challenges.

- The first challenge comes from the complexity of the brain. The brain is organized modularly, hierarchically, and across multi-scales (*Meunier et al., 2010*), implying that the framework must support model construction at different levels (e.g., channel, neuron, network) and model composition across multiple scales (e.g., neurons to networks, networks to circuits). Current brain simulators typically focus on only one or two scales, for example, spiking networks (*Gewaltig and Diesmann, 2007*; *Davison et al., 2008*; *Beyeler et al., 2015*; *Stimberg et al., 2019*) or firing rate models (*Sanz Leon et al., 2013*; *Cakan et al., 2021*). Recently, NetPyNE (*Dura-Bernal et al., 2019*) and BMTK (*Dai et al., 2020a*) have adopted descriptive languages to expand the modeling scales from channels to neurons and networks, but their modeling interfaces are still limited to predefined scales.
- The second challenge is the integration of different modeling needs (*Ramezanian-Panahi et al., 2022*; *D'Angelo and Jirsa, 2022*). To elucidate brain functions comprehensively with computational models, we need to not only simulate neural activities, but also analyze the underlying mechanisms, and sometimes, we need to train models from data or tasks, implying that a general-purpose programming framework needs to be a platform to integrate multiple modeling demands. Current brain simulators mainly focus on simulation (*Brette et al., 2007*; *Tikidji-Hamburyan et al., 2017*; *Blundell et al., 2018*), and largely ignore training and analysis.
- The third challenge is achieving high-running performance while maintaining programming convenience (*Tikidji-Hamburyan et al., 2017*; *Blundell et al., 2018*), which is particularly true for brain dynamics modeling, as its unique characteristics make it difficult to run efficiently within a convenient Python interface. The current popular approach for solving this challenge is code generation based on descriptive languages (*Goodman, 2010*; *Blundell et al., 2018*). However, this approach has intrinsic limitations regarding transparency, flexibility, and extensibility (*Tikidji-Hamburyan et al., 2017*; *Blundell et al., 2018*) (Appendix 1).
- The fourth challenge comes from the rapid development of the field. Brain dynamics modeling is relatively new and developing rapidly. New concepts, models, and mathematical approaches are constantly emerging, implying that a general-purpose programming framework needs to be extensible to take up new advances in the field conveniently.

In this paper, we propose BrainPy ('Brain Dynamics Programming in Python', *Figure 1*) as a solution to address all the above challenges. BrainPy provides infrastructure tailored for brain dynamics programming, including mathematical operators, differential equation solvers, universal model-building formats, and object-oriented JIT compilation. Such infrastructure provides the flexibility for users to define brain dynamics models freely and lays the foundation for BrainPy to build an integrative framework for brain dynamics modeling. First, BrainPy introduces a `brainpy.Dynamical-System` interface to unify diverse brain dynamics models. Models at any level of resolution can be defined as `DynamicalSystem` classes, which further can be hierarchically composed to create higher-level models. Second, BrainPy builds an integrated platform for studying brain dynamics models, where the same BrainPy model can be used for simulation, training (e.g., offline learning, online learning, or backpropagation training), and analysis (e.g., low-dimensional bifurcation analysis or high-dimensional slow point analysis). Third, through JIT compilation and dedicated operators, BrainPy achieves impressive performance for its code execution. The same models can be deployed into different devices (such as Central Processing Unit [CPU], Graphics Processing Unit [GPU], and Tensor Processing Unit [TPU]) without additional code modification. Fourth, BrainPy is highly extensible. New extensions can be easily implemented as plug-in modules. Even the low-level primitive operators in the kernel system can be extended in the user-level Python interface. BrainPy is implemented in a

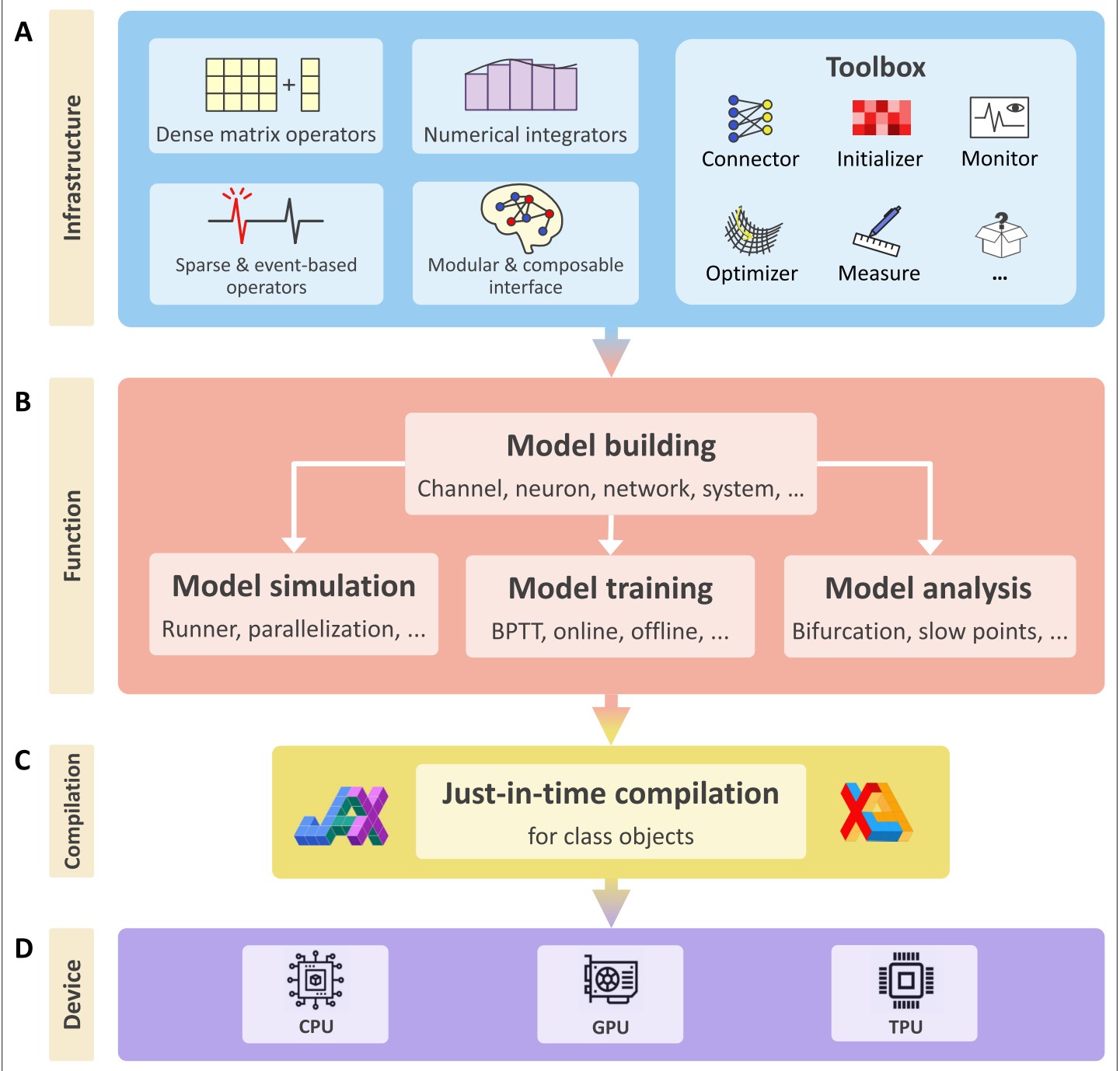

**Figure 1.** BrainPy is an integrative framework targeting general-purpose brain dynamics programming. (**A**) Infrastructure: BrainPy provides infrastructure tailored for brain dynamics programming, including NumPy-like operators for computations based on dense matrices, sparse and event-based operators for event-driven computations, numerical integrators for solving diverse differential equations, the modular and composable programming interface for universal model building, and a toolbox useful for brain dynamics modeling. (**B**) Function: BrainPy provides an integrated platform for studying brain dynamics, including model building, simulation, training, and analysis. Models defined in BrainPy can be used for simulation, training, and analysis jointly. (**C**) Compilation: Based on JAX (*Frostig et al., 2018*) and XLA (*Sabne, 2020*), BrainPy provides just-in-time (JIT) compilation for Python class objects. All models defined in BrainPy can be JIT compiled into machine codes to achieve high-running performance. (**D**) Device: The same BrainPy model can run on different devices including Central Processing Unit (CPU), Graphics Processing Unit (GPU), or Tensor Processing Unit (TPU), without additional code modification.

robust continuous integration pipeline and is equipped with an automatic documentation building environment (Appendix 3). It is open sourced at https://github.com/brainpy/BrainPy. Rich tutorials and extensive examples are available at https://brainpy.readthedocs.io and https://brainpy-examples. readthedocs.io, respectively.

## Method and results

### Infrastructure tailored for brain dynamics programming

To support its goal of becoming a general-purpose programming framework, BrainPy provides the infrastructure essential for brain dynamics modeling (*Figure 1A*). This infrastructure is a collection of interconnected utilities designed to provide foundational services that enable users to easily, flexibly, and efficiently perform various types of modeling for brain dynamics. Specifically, BrainPy implements (1) mathematical operators for conventional computation based on dense matrices and event-driven computation based on sparse connections; (2) numerical integrators for various differential equations, the backbone of dynamical neural models; (3) a universal model-building interface for constructing multi-scale brain dynamics models and the associated JIT compilation for the efficient running of these models; and (4) a toolbox specialized for brain dynamics modeling.

First, BrainPy delivers rich mathematical operators as essential elements to describe diverse brain dynamics models (Appendix 4). On the one hand, brain dynamics modeling involves conventional computation based on dense matrices. In Python scientific computing ecosystem, dense matrix operators have been standardized and popularized by NumPy (*Harris et al., 2020*), TensorFlow (*Abadi et al., 2016*), and PyTorch (*Paszke et al., 2019*). To reduce the cost of learning a new set of computing languages, dense matrix operators in BrainPy (including multi-dimensional arrays, mathematical operations, linear algebra routines, Fourier transforms, and random number generations) follow the syntax of those in NumPy, TensorFlow, and PyTorch so that most Python users can directly program in BrainPy with their familiar operator syntax. On the other hand, brain dynamics modeling has specific computation properties, such as sparse connections and event-driven computations, which are difficult to efficiently implement with conventional operators. To accommodate these needs, BrainPy provides dozens of dedicated operators tailored for brain dynamics modeling, including event-driven operators, sparse operators, and JIT connectivity operators. Compared to traditional dense matrix operators, these operators can reduce the running time of typical brain dynamics models by several orders of magnitude (see Efficient performance of BrainPy).

Second, BrainPy offers a repertoire of numerical solvers for solving differential equations (Appendix 5). Differential equations are involved in most brain dynamics models. For ease of use, BrainPy's numerical integration of differential equations is designed as a Python decorator. Users define differential equations as Python functions, whose numerical integration is accomplished by calling integrator functions, for example, `brainpy.odeint()` for ordinary differential equations (ODEs), `brainpy. sdeint()` for stochastic differential equations (SDEs), and `brainpy.fdeint()` for fractional differential equations (FDEs). These integrator functions are designed to be general, and most numerical solvers for ODEs and SDEs are provided, such as explicit Runge–Kutta methods, adaptive Runge–Kutta methods, and Exponential methods. For SDEs, BrainPy supports different stochastic integrals (Itô or Stratonovich) and different types of Wiener processes (scalar or multi-dimensional). As delays are ubiquitous in brain dynamics, BrainPy also supports the numerical integration of delayed ODEs, SDEs, and FDEs with various delay forms.

Third, BrainPy supports modular and composable programming and the associated object-oriented transformations (Appendix 6). To capture the fundamental characteristics of brain dynamics, which are modular, multi-scaled, and hierarchical (*Meunier et al., 2010*), BrainPy follows the philosophy that 'any dynamical model is just a Python class, and high-level models can be recursively composed by low-level ones' (details will be illustrated in Flexible model building in BrainPy). However, such a modular and composable interface is not directly compatible with JIT compilers such as JAX and Numba, because they are designed to work with pure functions (Appendix 2). By providing object-oriented transformations, including the JIT compilation for class objects and the automatic differentiation for class variables, models defined with the above modular and composable interface can also benefit from the powerful transformations in advanced JIT compilers.

Fourth, BrainPy offers a toolbox specialized for brain dynamics modeling. A typical modeling experiment involves multiple stages or processes, such as creating synaptic connectivity, initializing connection weights, presenting stimulus inputs, and analyzing simulated results. For the convenience of running these operations repeatedly, BrainPy presets a set of utility functions, including synaptic connection, weight initialization, input construction, and data analysis. However, this presetting does not prevent users from defining their utility functions in the toolbox.

## Flexible model building in BrainPy

Brain dynamics models have the key characteristics of being modular, multi-scaled, and hierarchical, and BrainPy designs a modular, composable, and flexible programming paradigm to match these features. The paradigm is realized by the DynamicalSystem interface, which has the following appealing features.

`DynamicalSystem` supports the definition of brain dynamics models at any organization level. Given a dynamical system, regardless of its complexity, users can implement it as a `Dynamical-System` class. As an example, *Figure 2A* demonstrates how to define a potassium channel model with `DynamicalSystem`, in which the initialization function defines parameters and states, and the update function specifies how the states evolve. In this process, BrainPy toolbox can help users quickly initialize model variables, synaptic connections, weights, and delays, and BrainPy operators and integrators can support users to define model updating logic freely. In a similar fashion, other dynamical models, such as discontinuous neuron models (e.g., leaky integrate-and-fire model; *Abbott, 1999*), continuous neuron models (e.g., FitzHugh–Nagumo model; *Fitzhugh, 1961*), population models (e.g., Wilson–Cowan model; *Wilson and Cowan, 1972*), and network models (e.g., continuous attractor neural network; *Wu et al., 2008*), can be implemented by subclassing DynamicalSystem as standalone modules.

However, for complex dynamical models, such as Hodgkin–Huxley (HH)-typed neuron models or large-scale cortical networks, their model definitions can be achieved through the composition of subcomponents. All models defined with `DynamicalSystem` can be used as modules to form more complicated high-level models. As an example, *Figure 2B* demonstrates how an HH-typed neuron model is created by combining multiple ion channel models. Such composable programming is the core of `DynamicalSystem`, and applies to almost all BrainPy models. For example, a synapse model consists of four components: synaptic dynamics (e.g., alpha, exponential, or dual exponential dynamics), synaptic communication (e.g., dense, sparse, or convolutional connections), synaptic output (e.g., conductance-, current-, or magnesium blocking-based), and synaptic plasticity (e.g., short- or long-term plasticity). Composing different realizations of these components enables to create diverse kinds of synaptic models. Similarly, various network models can be implemented by combining different neuron groups and their synaptic projections.

Remarkably, `DynamicalSystem` supports hierarchical composable programming, such that a model composed of lower-level components can hierarchically serve as a new component to form higher-level models. This property is highly useful for the construction of multi-scale brain models. *Figure 2* demonstrates an example of recursively composing a model from channels (*Figure 2A*) to neurons (*Figure 2B*) to networks (*Figure 2C*) and to systems (*Figure 2D*, see Appendix 9 for details of the full model). It is worth pointing out that this hierarchical composition property is not shared by other brain simulators, and BrainPy allows for flexible control of composition depth according to users' needs. Moreover, for user convenience, BrainPy provides dozens of commonly used models, including channels, neurons, synapses, populations, and networks, as building blocks to simplify the building of large-scale models.

## Integrated modeling in BrainPy

BrainPy offers an integrated platform to comprehensively perform simulation, training, and analysis of brain dynamics models.

### Model simulation

BrainPy designs the interface `brainpy.DSRunner` to simulate the dynamics of brain models. DSRunner can be used to simulate models at any level, including but not limited to channel (*Figure 3A*), neuron (*Figure 3B*), network (*Figure 3C*), and system (*Figure 3D*) levels.

**A**
```
import brainpy as bp
import brainpy.math as bm

class IK(bp.dyn.IonChannel):
  def __init__(self, size, E=-90, g_max=10):
    super().__init__(size)
    self.g_max, self.E = g_max, E  # parameters
    self.p = bm.Variable(bm.zeros(size))  # variables
    self.integral = bp.odeint(self.dp, 'exp_euler')

  def update(self, V):
    self.p.value = self.integral(self.p, bp.share['t'], V)
```

**B**
```
class HH(bp.dyn.CondNeuGroup):
  def __init__(self, size):
    super().__init__(size)
    self.IK = IK(size, g_max=30.)
    self.INa = bp.dyn.INa_TM1991(size, g_max=100.)
    self.IL = bp.dyn.IL(size, E=-60., g_max=0.05)
```

**C**
```
class Network(bp.DynSysGroup):
  def __init__(self, num_E, num_I):
    super().__init__()
    self.E, self.I = HH(num_E), HH(num_I)
    self.E2E = Exponential(self.E, self.E, 0.02,
                           g_max=0.03, tau=5, E=0.)
    self.E2I = Exponential(self.E, self.I, 0.02,
                           g_max=0.03, tau=5., E=0.)
    self.I2E = Exponential(self.I, self.E, 0.02,
                           g_max=0.335, tau=10., E=-80)
    self.I2I = Exponential(self.I, self.I, 0.02,
                           g_max=0.335, tau=10., E=-80.)
```

**D**
```
class System(bp.DynSysGroup):
  def __init__(self, conn, delay, num):
    super().__init__()
    self.areas = [Network(3200, 800) for _ in range(num)]
    self.conns = []
    for i in range(num):
      for j in range(num):
        if i != j:
          proj = Projection(self.areas[j], self.areas[i],
                            delay=delay[i, j], conn=conn[i, j])
          self.conns.append(proj)
```

**Figure 2.** BrainPy supports modular and composable programming for building hierarchical brain dynamics models. (**A**) An ion channel model is defined as a subclass of `brainpy.dynz.IonChannel`. The `__init__()` function specifies the parameters and states, while the update() function defines the updating rule for the states. (**B**) An Hodgkin–Huxley (HH)-typed neuron model is defined by combining multiple ion channel models as a subclass of `brainpy.dyn.CondNeuGroup`. (**C**) An E/I balanced network model is defined by combining two neuron populations and their connections as a subclass of `brainpy.DynSysGroup`. (**D**) A ventral visual system model is defined by combining several networks, including V1, V2, V4, TEo, and TEpd, as a subclass of `brainpy.DynSysGroup`. For detailed mathematical information about the complete model, please refer to Appendix 9.

Brain dynamics models often require intensive parameter searches to fit the experimental data, which is a computationally demanding task. BrainPy facilitates this process by supporting multiple parallel simulation methods. Firstly, the `brainpy.running` module offers convenient routines for concurrent executions based on the python multiprocessing mechanism. This method is flexible, but

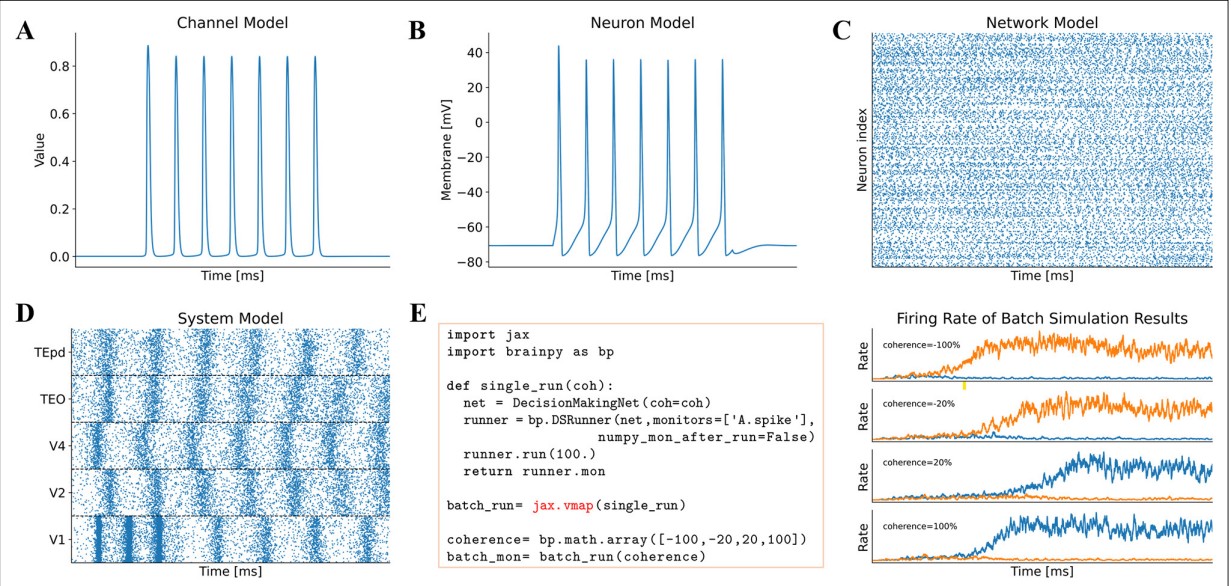

**Figure 3.** Model simulation in BrainPy. The interface `DSRunner` supports the simulation of brain dynamics models at various levels. (**A**) The simulation of the potassium channel in *Figure 2A*. (**B**) The simulation of the HH neuron model in *Figure 2B*. (**C**) The simulation of the E/I balanced network, COBAHH (*Brette et al., 2007*) in *Figure 2C*. (**D**) The simulation of the ventral visual system model (the code please see *Figure 2D*, and the model please see Appendix 9). (**E**) Using jax.vmap to run a batch of spiking decision-making models (*Wang, 2002*) with inputs of different coherence levels. The left panel shows the code used for batch simulations of different inputs, and the right panel illustrates the firing rates under different inputs.

may introduce additional time overhead due to the model recompilation and reinitialization in each process. Secondly, most BrainPy models inherently support the automatic vectorization of `jax.vmap` and automatic parallelization of `jax.pmap`. These methods can avoid the recompilation and reinitialization of models in the same batch, and automatically parallelize the model execution on the given machines. *Figure 3E* illustrates the simplicity of this batch simulation approach. By using a single line of functional calls, BrainPy models can run simultaneously with different parameter settings.

## Model training

The use of machine-learning methods to train neural models is becoming a new trend for studying brain functions (*Masse et al., 2019*; *Finkelstein et al., 2021*; *Laje and Buonomano, 2013*; *Sussillo et al., 2015*; *Saxe et al., 2021*). BrainPy provides the `brainpy.DSTrainer` interface to support this utility. Different subclasses of `DSTrainer` provide different training algorithms, which can be used to train different types of models. For instance, the trainer `brainpy.BPTT` implements the algorithm of backpropagation through time, which is helpful for training spiking neural networks (*Figure 4A*) and recurrent neural networks (*Figure 4B*). Similarly, `brainpy.OfflineTrainer` implements offline learning algorithms such as ridge regression (*Lukoševičius, 2012*), `brainpy.OnlineTrainer` implements online learning algorithms such as FORCE learning (*Sussillo and Abbott, 2009*), which are useful for training reservoir computing models (*Figure 4C*). In a typical training task, one may try different algorithms that can be used to train a model. The unified syntax for defining and training models in BrainPy enables users to train the same model using multiple algorithms (see Appendix 10). *Figure 4D–F* demonstrates that a reservoir network model can be trained with three different algorithms (online, offline, and backpropagation) to accomplish a classical task of chaotic time series prediction (*Jaeger, 2007*).

Since the training algorithms for brain dynamics models have not been standardized in the field, BrainPy provides interfaces to support the flexible customization of training algorithms. Specifically, `OfflineTrainer` and `OnlineTrainer` provide general interfaces for offline and online learning algorithms, respectively, and users can easily select the appropriate method by specifying the fit_method parameter in `OfflineTrainer` or `OnlineTrainer`. Furthermore, the BPTT interface is designed to capture the latest advances in backpropagation algorithms. For instance, it supports eligibility

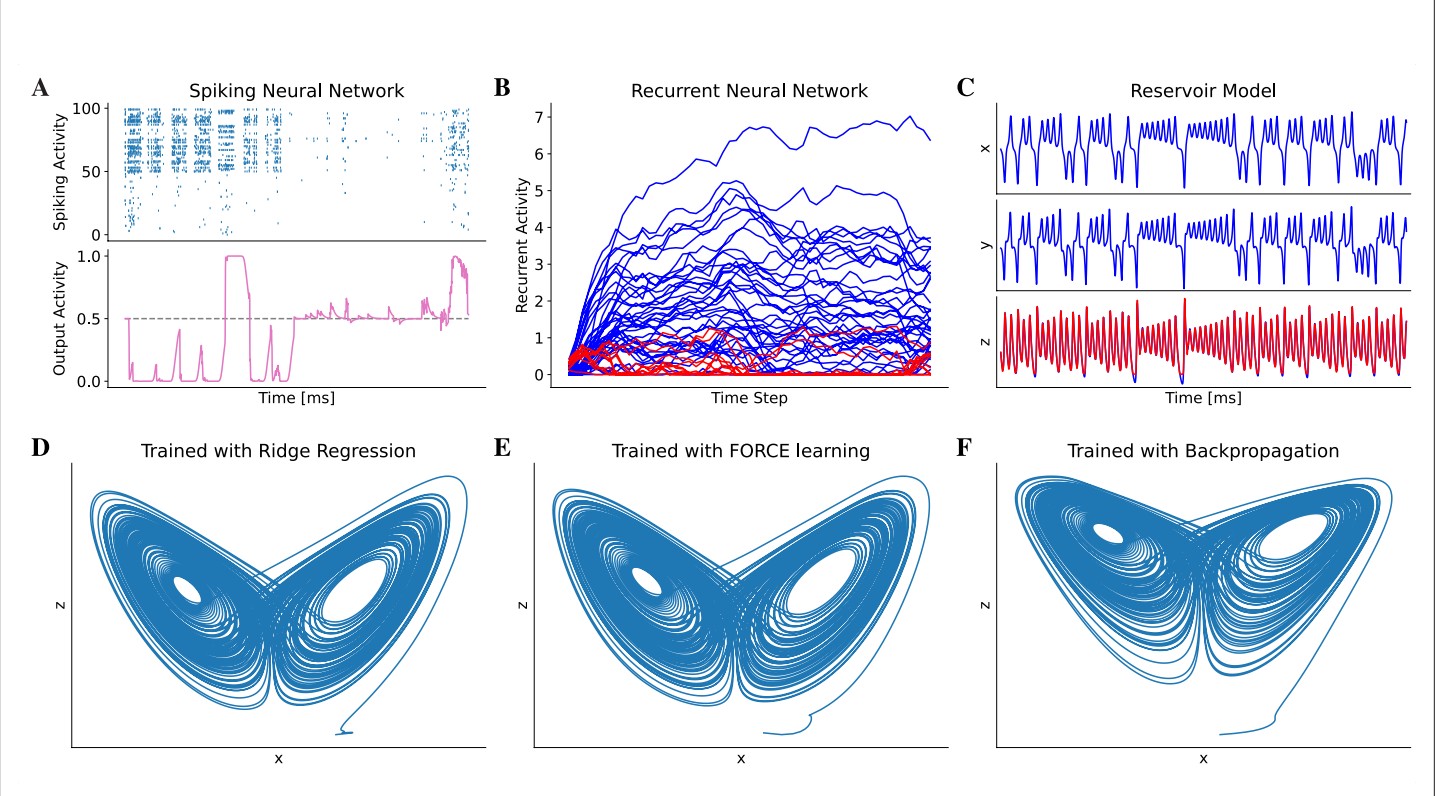

**Figure 4.** Model training in BrainPy. BrainPy supports the training of brain dynamics models from data or tasks. (**A**) Training a spiking neural network (*Bellec et al., 2020*) on an evidence accumulation task (*Morcos and Harvey, 2016*) using the backpropagation algorithm with `brainpy.BPTT`. (**B**) Training an artificial recurrent neural network model (*Song et al., 2016*) on a perceptual decision-making task (*Britten et al., 1992*) with `brainpy.BPTT`. (**C**) Training a reservoir computing model (*Gauthier et al., 2021*) to infer the Lorenz dynamics with the ridge regression algorithm implemented in `brainpy.OfflineTrainer`. $x, y$, and $z$ are variables in the Lorenz system. (**D–F**) The classical echo state machine (*Jaeger, 2007*) has been trained using multiple algorithms to predict the chaotic Lorenz dynamics. The algorithms utilized include ridge regression (**D**), FORCE learning (**E**), and backpropagation algorithms (**F**) implemented in `BrainPy`. The mean squared errors between the predicted and actual Lorenz dynamics were 0.001057 for ridge regression, 0.171304 for FORCE learning, and 1.276112 for backpropagation. Please refer to Appendix 10 for the training details.

propagation algorithm (*Bellec et al., 2020*) and surrogate gradient learning (*Neftci et al., 2019*) for training spiking neural networks.

## Model analysis

Analyzing model dynamics is as essential as model simulation and training because it helps unveil the underlying mechanism of model behaviors. Given a dynamical system, BrainPy provides the interface `brainpy.DSAnalyzer` for automatic dynamic analysis, and different classes of DSAnalyzer implement different analytical methods.

First, BrainPy supports phase plane and bifurcation analyses for low-dimensional dynamical systems. The phase plane is a classical and powerful technique for the analysis of dynamical systems and has been widely used in brain dynamics studies, including neuron models (e.g., Izhikevich model; *Izhikevich, 2003*) and population rate models (e.g., Wilson–Cowan model; *Wilson and Cowan, 1972*). *Figure 5A* shows an example where many features of phase plane analysis, including nullcline, vector field, fixed points, and their stability, for a complex rate-based decision-making model (*Wong and Wang, 2006*) are automatically evaluated by several lines of BrainPy code. Bifurcation analysis is another utility of BrainPy, which allows users to easily investigate the changing behaviors of a dynamical system when parameters are continuously varying. *Figure 5B* demonstrates the stability changes of the classical FitzHugh–Nagumo model (*Fitzhugh, 1961*) with one parameter varying can be easily inspected by the bifurcation analysis interface provided in BrainPy. Similarly, bifurcation analysis of codimension-2 (with two parameters changing simultaneously; *Figure 5C*) can be performed with the

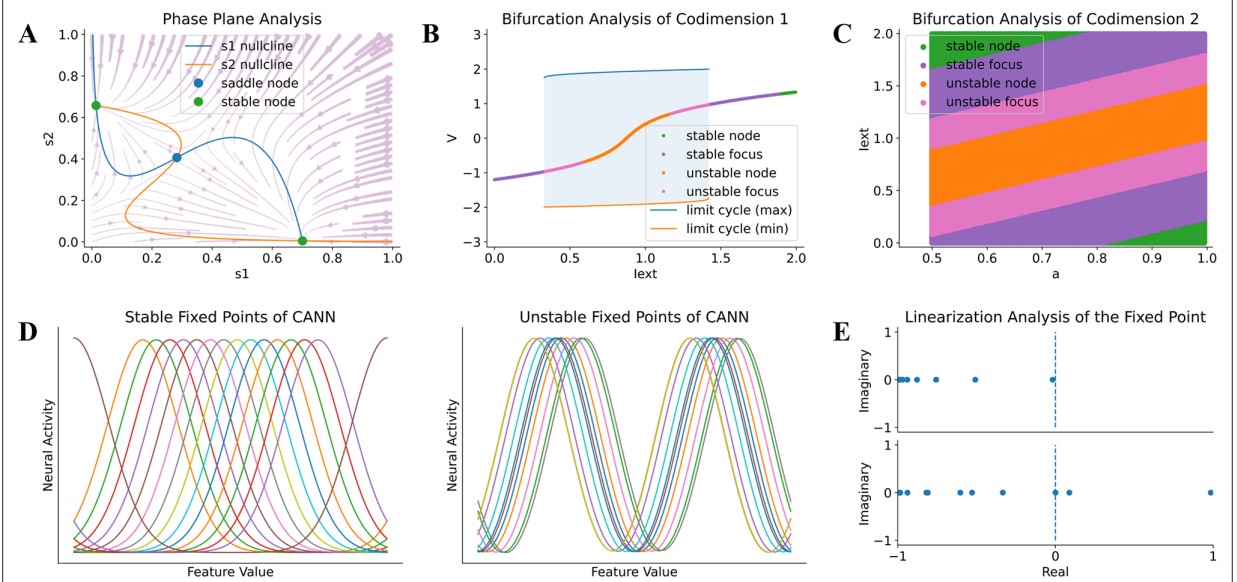

**Figure 5.** Model analysis in BrainPy. BrainPy supports automatic dynamics analysis for low- and high-dimensional systems. (**A**) Phase plane analysis of a rate-based decision-making model (**Wong and Wang, 2006**). (**B**) Bifurcation analysis of codimension 1 of the FitzHugh–Nagumo model (**Fitzhugh, 1961**), in which the bifurcation parameter is the external input `Iext`. (**C**) Bifurcation analysis of codimension 2 of the FitzHugh–Nagumo model (**Fitzhugh, 1961**), in which two bifurcation parameters `Iext` and `a` are continuously varying. (**D**) Finding stable and unstable fixed points of a high-dimensional CANN model (**Wu et al., 2008**). (**E**) Linearization analysis of the high-dimensional CANN model (**Wu et al., 2008**) around one stable and one unstable fixed point.

same interface. BrainPy also supports bifurcation analysis for three-dimensional fast–slow systems, for example, a bursting neuron model (**Rinzel, 1985**). This set of low-dimensional analyzers is performed numerically so that they are not restricted to equations with smooth functions, but are equally applicable to ones with strong and complex nonlinearity.

Second, BrainPy supports slow point computation and linearization analysis for high-dimensional dynamical systems. With powerful numerical optimization methods, one can find fixed or slow points of a high-dimensional nonlinear system (**Sussillo and Barak, 2013**). By integrating numerical methods such as gradient descent and nonlinear optimization algorithms, BrainPy provides the interface `brainpy.analysis.SlowPointFinder` as a fundamental tool for high-dimensional analysis. **Figure 5D** demonstrates that the `SlowPointFinder` can effectively find a line of stable and unstable attractors in a CANN network (**Wu et al., 2008**). Furthermore, the linearized dynamics around the found fixed points can be easily inspected and visualized with `SlowPointFinder` interface (**Figure 5E**).

## Efficient performance of BrainPy

Simulating dynamical models efficiently in Python is notoriously challenging (**Blundell et al., 2018**). To resolve this problem, BrainPy leverages the JIT compilation of JAX/XLA and exploits dedicated primitive operators to accelerate the model running.

### JIT compilation

In contrast to deep neural networks (DNNs), which mainly consist of computation-intensive operations (such as convolution and matrix multiplication), brain dynamics models are usually dominated by memory-intensive operations. Taking the classical leaky integrate-and-fire (LIF) neuron model (**Abbott, 1999**) as an example, its computation mainly relies on operators such as addition, multiplication, and division. As shown in **Figure 6A**, we measure the running times of an LIF model and a matrix multiplication with the same number of floating-point operations (FLOPs) on both CPU and GPU devices. The results indicate that the LIF model is significantly slower than the matrix multiplication on both devices, despite having the same theoretical complexity. This reveals the existence of large overheads when executing brain dynamics models in Python. Moreover, these overheads become dominant

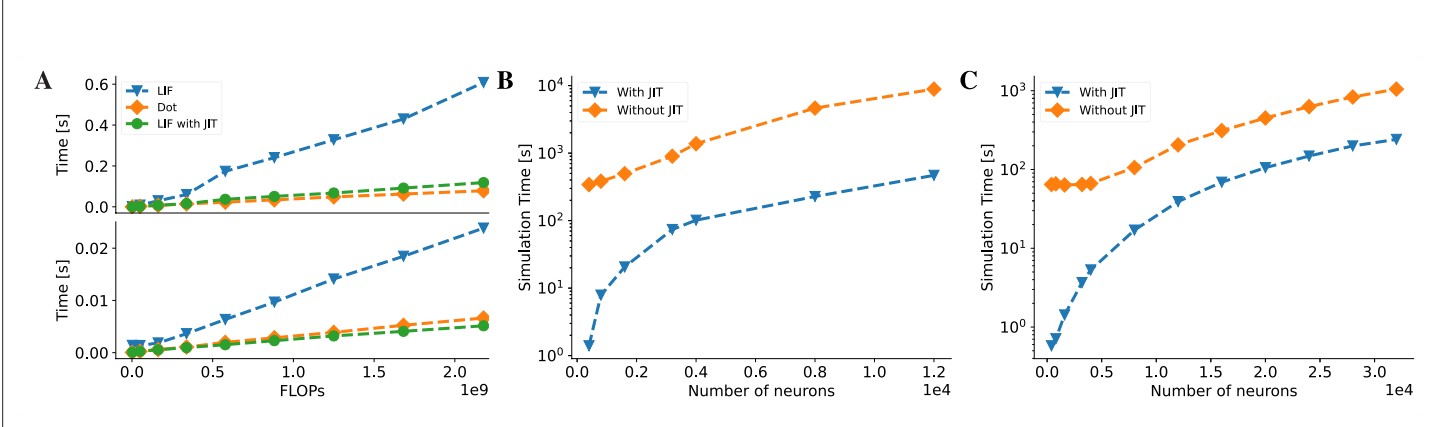

**Figure 6.** BrainPy accelerates the running speed of brain dynamics models through just-in-time (JIT) compilation. (**A**) Performance comparison between an LIF neuron model (***Abbott, 1999***) and a matrix–vector multiplication $Wv$ ($W \in \mathbb{R}^{m \times m}$ and $v \in \mathbb{R}^{m}$). By adjusting the number of LIF neurons in a network and the dimension $m$ in the matrix–vector multiplication, we compare two models under the same floating-point operations (FLOPs). The top panel: On the Central Processing Unit (CPU) device, the LIF model without JIT compilation (the 'LIF' line) shows much slower performance than the matrix–vector multiplication (the 'Dot' line). After compiling the whole LIF network into the CPU device through JIT compilation (the 'LIF with JIT' line), two models show comparable running speeds (please refer to ***Appendix 11—figure 6A*** for the time ratio). The bottom panel: On the Graphics Processing Unit (GPU) device, the LIF model without JIT shows several times slower than the matrix–vector multiplication under the same FLOPs. After applying the JIT compilation, the jitted LIF model shows comparable performance to the matrix–vector multiplication (please refer to ***Appendix 11—figure 6B*** for the time ratio). (**B, C**) Performance comparison of a classical E/I balanced network COBA (***Vogels and Abbott, 2005***) with and without JIT compilation (the 'With JIT' line vs. the 'Without JIT' line). (**B**) JIT compilation provides a speedup of over ten times for the COBA network on the CPU device (please refer to ***Appendix 11—figure 6C*** for the acceleration ratio). (**C**) Similarly, after compiling the whole COBA network model into GPUs, the model achieves significant acceleration, several times faster than before (please refer to ***Appendix 11—figure 6D*** for the acceleration ratio). For experimental details, please see Appendix 11.

when simulating large-scale brain networks, as they grow rapidly with the number of operators in the model.

To overcome this limitation, we employ the JIT compilation technique to dramatically reduce these overhead costs in BrainPy. The JIT compilation transforms the dynamic Python code into the static machine code during runtime, which can significantly reduce the time cost of Python interpretation. Specifically, we utilize JAX, which implements JIT compilation based on XLA (Appendix 2). The XLA JIT engine employs specialized optimizations for memory-intensive operators, for example, operator fusion, which alleviates memory access overhead by minimizing the requirement for intermediate data storage and redundant data transfers during the sequential execution of multiple unmerged operations. This renders the JIT compilation with XLA highly suitable for handling brain dynamics models. ***Figure 6A*** demonstrates that with the JIT compilation, the LIF model achieves a running speed comparable to that of the matrix multiplication operation Dot on the CPU and outperforms or matches it on the GPU (see ***Figure 6A***, ***Appendix 11—figure 6A***, and ***Appendix 11—figure 6B***). To further illustrate the benefits of the JIT compilation, we apply it to a realistic brain simulation model, namely, the E/I balanced network model COBA (***Vogels and Abbott, 2005***). The results show that the JIT compilation boosts the running speed by 10 times on both the CPU and GPU compared to the case without JIT compilation (for CPU acceleration, see ***Figure 6B*** and ***Appendix 11—figure 6C***; for GPU acceleration, see ***Figure 6C*** and ***Appendix 11—figure 6D***).

## Dedicated operators

Another key feature that distinguishes brain dynamics models from DNNs is that they usually have sparse connections and perform event-driven computations. For example, neurons in a network are typically connected with each other with a probability less than 0.2 (***Potjans and Diesmann, 2014***), and the state of a postsynaptic neuron is updated only when a presynaptic spike event occurs. These unique features greatly impair the efficiency of brain model simulation using conventional operators, even with the help of JIT compilation. To illustrate this, ***Figure 7A*** demonstrates that when implementing a COBA network model with dense matrix-based operators, the majority of simulation time

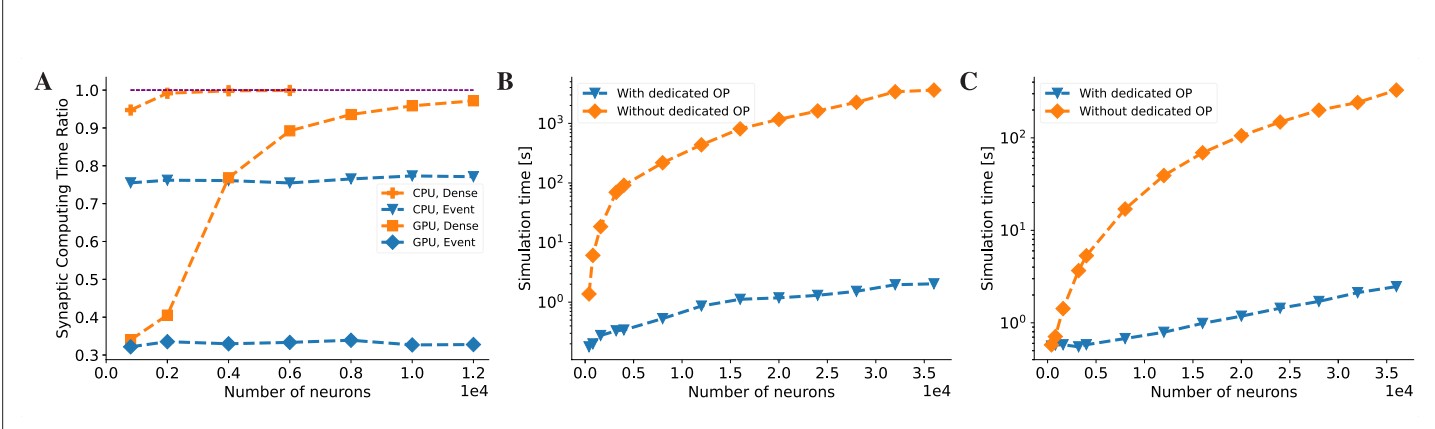

**Figure 7.** BrainPy accelerates the running speed of brain dynamics models through dedicated operators. (**A**) Without dedicated event-driven operators, the majority of the time is spent on synaptic computations when simulating a COBA network model (***Vogels and Abbott, 2005***). The ratio significantly increases with the network size on both Central Processing Unit (CPU) and Graphics Processing Unit (GPU) devices (please refer to the lines labeled as '`CPU, Dense`' and '`GPU, Dense`' which correspond to the models utilizing the dense operator-based synaptic computation and running on the CPU and GPU devices, respectively). With the event-based primitive operators, the proportion of time spent on synaptic computation remains constant regardless of network size (please refer to the lines labeled as '`CPU, Event`' and '`GPU, Event`' which represent the models performing event-driven computations on the CPU and GPU devices, respectively). (**B**) On the CPU device, the COBA network model with event-based operators (see the '`With dedicated OP`' line) is accelerated by up to three orders of magnitude compared to that without dedicated operators (see the '`Without dedicated OP`' line). Please refer to ***Appendix 11—figure 7A*** for the acceleration ratio. (**C**) The COBA network model exhibited two orders of magnitude acceleration when implemented with event-based primitive operators on a GPU device. This performance improvement was more pronounced for larger network sizes on both CPU and GPU platforms. Please refer to ***Appendix 11—figure 7B*** for the acceleration ratio. For experimental details, please see Appendix 11.

is consumed by synaptic computations on both CPU and GPU devices, and this issue becomes more pronounced as the network size increases (see '`CPU, Dense`' and '`GPU, Dense`' lines in ***Figure 7A***).

In order to address this challenge, BrainPy introduces specialized primitive operators designed to accelerate event-based computations within sparsely connected networks. These specialized operators encompass transformations among variables associated with presynaptic neurons, postsynaptic neurons, and synapses, as well as sparse computation operators, event-driven computation operators, and JIT connectivity operators (refer to Appendix 4 for more details). By employing these specialized operators, BrainPy significantly reduces the time required for synaptic computations. As depicted in ***Figure 7B***, the specialized event-based operators result in a remarkable speedup of the classical COBA network model by orders of magnitude (see ***Appendix 11—figure 7A***). Similar speed improvements are observed when utilizing GPU computations, as shown in ***Figure 7C*** and ***Appendix 11—figure 7B***. Furthermore, an examination of the time proportion for synaptic computations indicates that the utilization of specialized operators ensures a consistent time ratio for synaptic computation, even as the network size increases (see '`CPU, Event`' and '`GPU, Event`' lines in ***Figure 7A***).

## Benchmarking

To conduct a formal assessment of the running efficiency of BrainPy, we conducted a comparative analysis against several widely used brain simulators, namely NEURON (***Hines and Carnevale, 1997***), NEST (***Gewaltig and Diesmann, 2007***), Brian2 (***Stimberg et al., 2019***), Brian2CUDA (***Alevi et al., 2022***), GeNN (***Yavuz et al., 2016***), and Brian2GeNN (***Stimberg et al., 2020***). Our benchmarking focused on measuring the simulation speeds of these frameworks for models with sparse and dense connectivity patterns. The tests were performed using three common computing platforms: CPU, GPU, and TPU. This comprehensive assessment provides insights into BrainPy's efficiency relative to other mainstream simulators across different hardware configurations and network scales.

To evaluate the performance of brain simulators on sparsely connected networks, we utilized two established E/I balanced network models with LIF and HH neuron types: the COBA (***Vogels and Abbott, 2005***) and COBAHH (***Brette et al., 2007***) networks (experimental details please see Appendix 11). COBA consists of excitatory and inhibitory LIF neurons with sparse random connectivity.

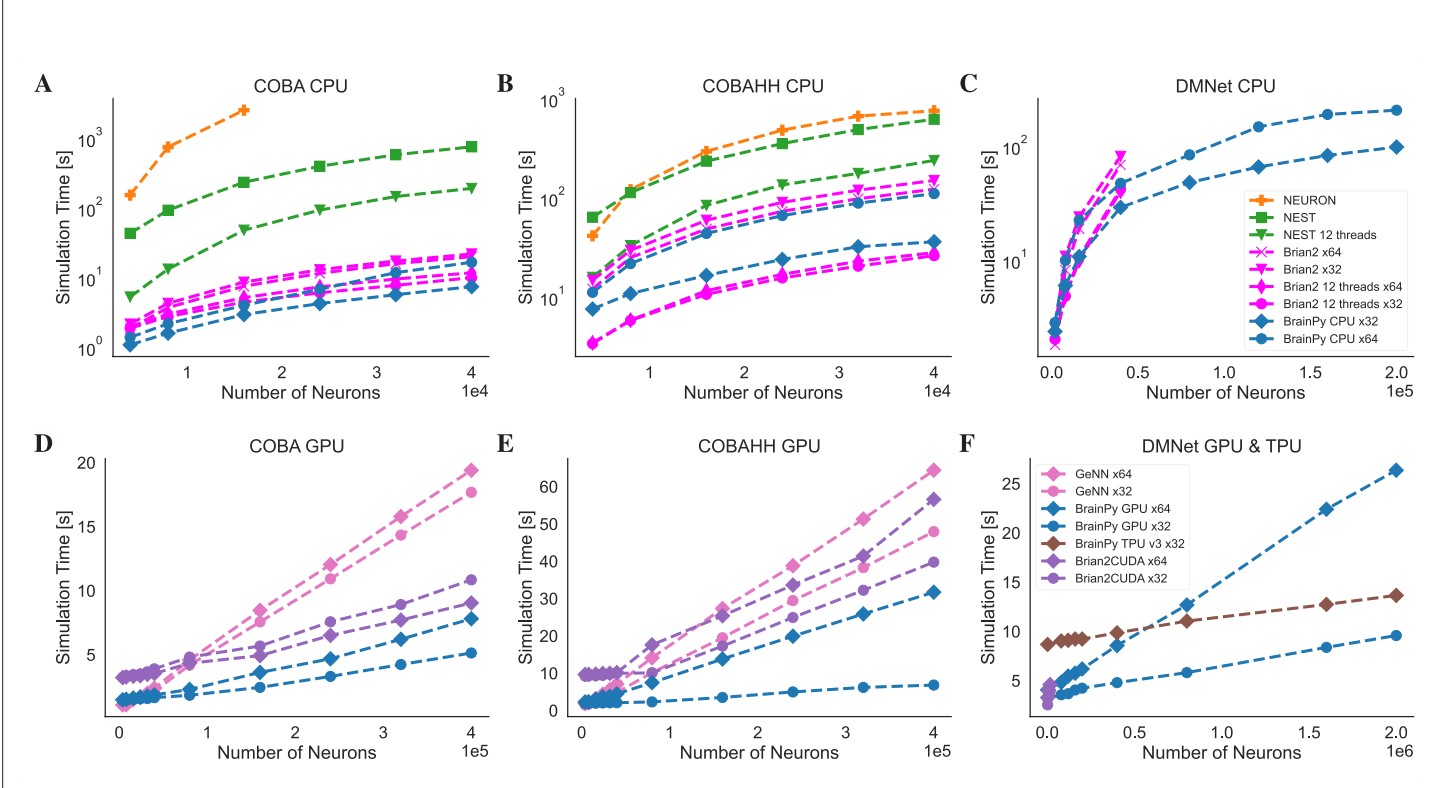

**Figure 8.** Speed comparison of NEURON, Nest, Brian2, and BrainPy under different computing devices. Comparing speeds of different brain simulation platforms using the benchmark model COBA (*Vogels and Abbott, 2005*) on both the Central Processing Unit (CPU) (**A**) and Graphics Processing Unit (GPU) (**D**) devices. NEURON is truncated at 16,000 neurons due to its very slow runtime. Comparing speeds of different platforms using the benchmark model COBAHH (*Brette et al., 2007*) on both the CPU (**B**) and GPU (**E**) devices. Speed comparison of a spiking decision-making network (*Wang, 2002*) on CPU (**C**), GPU, and Tensor Processing Unit (TPU) (**F**) devices. Please refer to Appendix 11 for experimental details, and *Appendix 11—figure 8* for more data.

COBAHH uses the same network architecture but replaces the LIF neurons with biophysically detailed HH neuron models. On the CPU platform, consistent with previous benchmark experiments (*Stimberg et al., 2019*), we find that NEURON and NEST simulators exhibit suboptimal performance when running on a single node (see *Figure 8A* and *Figure 8B*). In contrast, BrainPy and Brian2 demonstrate comparable performance, showcasing a remarkable speed advantage of one to two orders of magnitude over NEURON and NEST. As both Brian2 and BrainPy support single-precision floating-point computation (x32), we conducted an analysis of their performances in the context of x32 computation. In order to ensure accurate simulation results with x32 computation, we examined the simulation outcomes across various simulators and platforms (refer to Appendix 11). Our evaluation demonstrated that BrainPy outperforms Brian2 in terms of speedup for numerical integration using x32 arithmetic on CPU platforms. On the GPU platform, GeNN demonstrates optimal linear scaling of execution time on both COBA and COBAHH network models as the network size increases (*Figure 8D* and *Figure 8E*). In contrast, BrainPy and Brian2CUDA exhibit a slight overhead and maintain a constant running time when dealing with small network sizes. However, when it comes to network scaling, BrainPy and Brian2CUDA outperform GeNN. Particularly as the network size grows, GeNN exhibits significantly slower performance. Additionally, the utilization of single-precision floating point in GeNN, Brian2CUDA, and BrainPy further enhances their GPU performance (excluding the COBA model in Brian2CUDA). Once again, we observed that BrainPy's x32 mode achieves a more pronounced performance gain. Particularly, in the COBAHH model, BrainPy's x32 computation demonstrates a substantial speedup compared to other brain simulators. BrainPy also enables model deployment on TPUs. However, since TPUs currently lack native support for sparse computations and toolchains for operator customization, we could not leverage event-driven sparse operators to simulate the sparsely connected COBA and COBAHH networks. Instead, we used dense matrix multiplication with masking to approximate the

sparse connectivity. Unfortunately, this led to significantly slower performance for the two sparsely connected models compared to the results obtained on GPUs (please refer to *Appendix 11—figure 8*). Moreover, the use of masked matrices resulted in a quadratic increase in memory usage. Consequently, the benchmarking experiments of COBA and COBAHH networks on TPU were limited to a scale of $4e^4$ neurons.

To evaluate the performance of brain simulators on densely connected networks, we utilized the decision-making network proposed by *Wang, 2002*. Assessing computational efficiency for dense connectivity is important for simulating models that feature dense recurrent connections (*Motta et al., 2019*) and facilitating the integration with DNNs which commonly employ dense connectivity between layers (*Tavanaei et al., 2019*). Due to the considerably slower speeds observed and the absence of a publicly available implementation of a decision-making network model using NEURON and NEST, we have excluded them from this benchmark test. Additionally, we did not include a comparison with GeNN because Brian2GeNN does not support the translation of the advanced Brian2 feature employed in this model. Our evaluation showcases that Brian2, Brian2CUDA, and BrainPy exhibit comparable performance on networks of small sizes. However, BrainPy demonstrated substantially better scalability on larger network sizes (see *Figure 8C* and *Figure 8F*). For these types of simulation workloads with dense connectivity, TPUs significantly outperformed CPUs and GPUs. Since TPUs primarily utilize low-precision floating point (especially floating point with 16 bits) and are less optimized for double precision, we only tested the model with single-precision operations. Our evaluations clearly showcase the excellent scalability of the network as the size increases (refer to the GPU and TPU comparison in *Figure 8F*).

## Extensible architecture of BrainPy

Brain science, as well as brain dynamics modeling, is progressing rapidly. Along with the gigantic projects on brain research worldwide, new data and knowledge about brain structures and functions are constantly emerging, which impose new demands on brain dynamics modeling frequently, including, for instance, the simulation and analysis of large-size neural circuits, and the training of neural models based on recorded neural data. To be a general-purpose brain dynamics programming framework, the architecture of the framework must be extensible to conveniently take up new advances in the field. Current brain simulators based on descriptive languages have difficulty achieving this goal, since the extension of a new component through the descriptive interface needs to be done in both high- and low-level programming languages (Appendix 1). Through the elaborate architecture design, BrainPy enables easy extension with new infrastructure, new utility functions, and new machine-learning methods, all performed in our convenient Python interface.

First, for infrastructure (*Figure 1A*), BrainPy provides a convenient way of customizing a new tool by defining a new subclass. For example, a new Runge–Kutta integrator can be created by inheriting from `brainpy.ode.ExplicitRKIntegrator` and specifying the Butcher tableau; a new connector can be implemented by deriving from `brainpy.conn.TwoEndConnector` and overriding initialization function and connection building function (see Appendix 7 for details). Since models and modeling methods have not yet been standardized in the field, the abstraction and summarization of primitive operators for brain dynamics modeling are largely lacking. Although BrainPy has provided dozens of dedicated operators, it would be too soon to establish a complete operator library for brain dynamics modeling. To simplify the process of operator customization, BrainPy provides the `brainpy.math.CustomOpByNumba` interface that allows users to write and register an operator directly with Python syntax. Specifically, to customize a primitive operator, users need to subclass `CustomOpByNumba` and implement two Python functions: the abstract evaluation function `eval_shape()` and concrete computation function `con_compute()` (see Appendix 8 for more information). Notably, this approach differs from the operator customization in most DNN frameworks, in which low-level operators must be implemented through C++ code. We confirmed that operators customized through the BrainPy interface have comparable and even better performance than those written in C++ (please refer to *Figure 9* for the results and Appendix 8 for the source code for comparison).

Second, for functional modules (*Figure 1B*), BrainPy enables an extension of a new module with BrainPy infrastructure, as the latter can be arbitrarily fused, chained, or combined to create new functions. For example, an analysis toolkit can be customized with BrainPy operators. Moreover, all

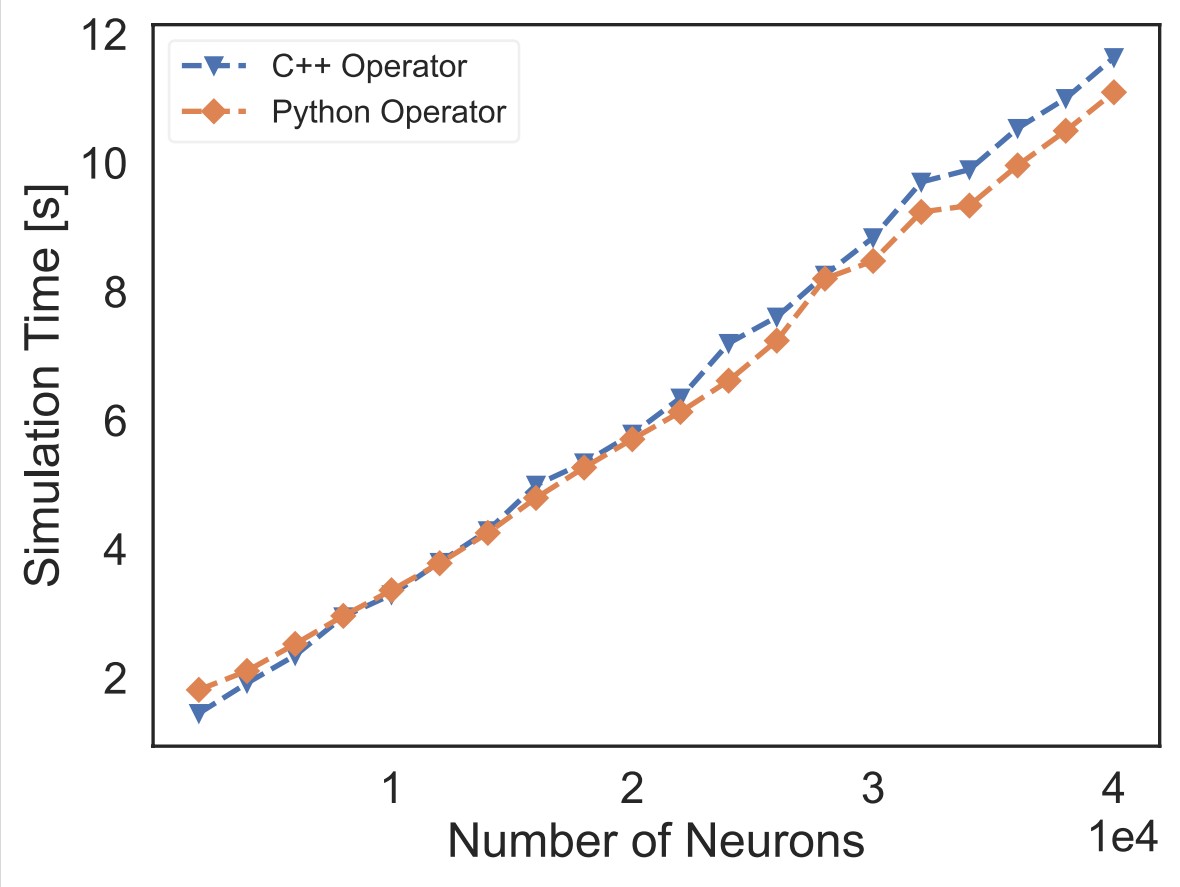

**Figure 9.** The speed comparison of event-based operators customized by C++ XLA custom call and our Python-level registration interface. 'C++ Operator' presents the simulation time of a COBA network using the event-based operator coded by C++, and 'Python Operator' shows the simulation speed of the network that is implemented through our operator registered by the Python interface.

customizations in BrainPy can benefit from the acceleration of JIT compilation, and users' attention only needs to focus on the functionalities they require.

Third, for interactions with AI, BrainPy supports the easy extension of new machine-learning methods. Machine-learning approaches are becoming important tools for brain dynamics modeling (*Saxe et al., 2021*). Existing brain simulators have difficulty incorporating the latest advances in machine-learning research (Appendix 1). Built on top of JAX, BrainPy has the inherent advantage of being linked to the latest developments in machine learning. We noticed that JAX has a rich ecosystem of machine learning, including DNNs, graph neural networks, reinforcement learning, and probabilistic programming. To integrate this rich ecosystem as part of the users' program, BrainPy is designed to be compatible with other JAX libraries. First, the object-oriented transformations in BrainPy can be applied to pure functions, thus enabling most JAX libraries with a functional programming style to be directly used as a part of the BrainPy program. Second, users can transform models in other libraries as BrainPy objects. For example, using `brainpy.dnn.FromFlax`, users can treat any artificial neural network model in Flax (*Heek, 2020*) as a BrainPy module. Alternatively, users can convert a BrainPy model into a format that is compatible with other JAX libraries. For instance, `brainpy.dnn.ToFlax` supports interpreting a dynamical system in BrainPy as a Flax recurrent cell, so that brain models in BrainPy can also be used in a machine-learning context.

## Discussion

The field of brain dynamics modeling has long been constrained by a lack of general-purpose programming frameworks that can support users to freely define brain dynamics models across multiple scales, comprehensively perform simulation, optimization, and analysis of the built models,

and conveniently prototype new modeling methods. To address this challenge, we have developed BrainPy, a general-purpose programming framework for brain dynamics modeling. With a combined focus on usability, performance, functionality, and extensibility, BrainPy offers a number of appealing properties, including:

- *Pythonic programming*. In contrast to other brain simulators (*Gewaltig and Diesmann, 2007*; *Davison et al., 2008*; *Beyeler et al., 2015*; *Stimberg et al., 2019*; *Hines and Carnevale, 1997*; *Dura-Bernal et al., 2019*; *Dai et al., 2020a*; *Goodman, 2010*; *Blundell et al., 2018*; *Tikidji-Hamburyan et al., 2017*), BrainPy enables Pythonic programming. It allows users to implement and control their models directly using native Python syntax, implicating high transparency to users. This transparency is crucial for research, as standard Python debugging tools can be integrated into the implementation process of novel models, and is also appealing for education.
- *Integrative platform*. BrainPy allows unprecedentedly integrated studying of brain dynamics models. Its multi-scale model-building interface facilitates the construction of data-driven models based on the structural, functional, or cellular data (*Potjans and Diesmann, 2014*), while its diverse model training supports enable to training brain dynamics models based on cognitive tasks that can be used to evaluate or optimize models of different brain functions (*Saxe et al., 2021*). BrainPy provides the first step toward an integrative framework supporting comprehensive brain modeling across different organization levels and problem dimensions (*D'Angelo and Jirsa, 2022*).
- *Intrinsic flexibility*. Inspired by the success of general-purpose programming in Deep Learning (*Abadi et al., 2016*; *Paszke et al., 2019*), BrainPy provides not only functional libraries but also infrastructure. This is essential for users to create models and modeling approaches beyond the predefined assumptions of existing libraries.
- *Efficient performance*. One of the key strengths of BrainPy lies in its ability to compile models defined in the framework into binary instructions for various devices, including CPU, GPU, and TPU. This compilation process ensures high-running performance comparable to native C or CUDA, enabling researchers to efficiently execute their models.
- *Extensible architecture*. BrainPy features an extensible architecture. New primitive operators, utilities, functional modules, machine-learning approaches, etc., can be easily customized through our Python interface.

## Limitations

While BrainPy's native Python-based object-oriented programming paradigm confers numerous advantages compared to existing brain simulators, this novel programming approach also imposes certain limitations that must be acknowledged.

Most existing brain simulators employ a domain-specific language to define brain dynamics models. For example, Brian2 (*Stimberg et al., 2019*) designs an equation-oriented specification that can describe a wide variety of neural models; NeuroML (*Cannon et al., 2014*) employs an XML-based specification that facilitates the sharing and reuse of neuronal models; NetPyNE (*Dura-Bernal et al., 2019*) utilizes a high-level JSON-compatible format composed of Python lists and dictionaries to support multi-scale neuronal modeling; BMTK (*Dai et al., 2020a*) similarly employs a JSON-based language built on the SONATA file format (*Dai et al., 2020b*) to deliver consistent multi-resolution experiences via integration with established tools like NEURON and NEST. This declarative programming approach benefits from a clear separation between the mathematical model description and its computational realization. It frees users from low-level implementation details, and enables intuitive specification of complex models in a concise and semantically clear manner. In contrast, the object-oriented programming used in BrainPy exposes the implementation details to users, and adds some complexity to the code. For example, users should be aware of the differences between dense and sparse connectivity schemes, online or offline training schemes, nonbatch or batch computing modes, etc.

The current objectives of BrainPy center on enabling an integrative platform for simulating, training, and analyzing large-scale brain network models while retaining biologically relevant details. Incorporating excessive biological details would be extremely computationally expensive and difficult for such integration. Consequently, detailed spatial modeling with complex compartmental dynamics, as facilitated by tools like NEURON (*Hines and Carnevale, 1997*) and Arbor (*Akar et al., 2019*), exceeds BrainPy's present scope. Moreover, in order to solve the governing partial differential equations, implicit numerical methods (e.g., Crank–Nicolson, implicit Euler) are often essential for stable

multi-compartment model simulation. As BrainPy does not currently support fully implicit solvers, it is ill suited to the needs and preferences of modelers focused on multi-compartment dynamics in its current form. Our emphasis remains on balancing biological fidelity and computational tractability for large-scale network modeling and training.

Based on the GSPMD mechanism of the XLA compiler (*Xu et al., 2021*), the current version of BrainPy supports various parallelism paradigms, such as data parallelism and model parallelism. Data parallelism involves dividing the training data across multiple devices, where each device independently computes and updates the model parameters using its assigned data subset. On the other hand, model parallelism entails partitioning the model across multiple devices, with each device responsible for computing a specific portion of the model computations. These parallelism paradigms are particularly applicable to brain dynamics models with dense connections or structured sparsity. However, the GSPMD parallelism mechanism is not straightforwardly applicable to sparse spiking neural networks, and requires non-trivial changes to support sparse computations. Therefore, another limitation of the current BrainPy framework is that it does not support the general parallelization of sparse spiking neural network models on multiple computing devices. State-of-the-art brain simulators now offer powerful parallelization capabilities for simulating large-scale SNNs. For instance, NEST (*Gewaltig and Diesmann, 2007*) and NEURON (*Hines and Carnevale, 1997*) simulators provide convenient and efficient commands through the MPI interface to define, connect, and execute large-scale networks. However, the array-based data structure in BrainPy requires a different approach to parallelize spiking neural networks.

## Future works

Although BrainPy offers substantial capabilities for brain dynamics modeling, fulfilling all demands in this domain will require large efforts for further ecosystem development.

First, supporting the efficient implementation of multi-compartment neuron models is needed to enable biologically detailed modeling at the subcellular level (*Poirazi and Papoutsi, 2020*). Multi-compartment neurons incorporate complex dendritic morphologies and spatially distributed ion channels that more precisely capture neural information processing. A substantial number of studies have demonstrated that dendritic mechanisms convey significant advantages to simplified network models of varying levels of abstraction (*Bono and Clopath, 2017*; *Legenstein and Maass, 2011*; *Wu et al., 2018*). Efficiently implementing such models in BrainPy could significantly advance detailed biophysical modeling and bridge the machine-learning-oriented SNN models.

Second, developing parallel primitive operators and memory-efficient algorithms will be critical for ultra-large-scale brain simulations approaching biological realism (>billions of neurons). Massive parallelization across multiple computing devices is currently the main approach to achieve such scale. For instance, the NEST simulator uses optimized data structures and algorithms (*Kunkel et al., 2011*; *Kunkel et al., 2014*; *Jordan et al., 2018*) to enable large-scale simulation on supercomputers and clusters. Moving forward, a priority for BrainPy will be parallelizing its array-based data structures to simulate gigantic brain models across multiple nodes. Moreover, rather than solving large-scale networks exactly, BrainPy aims to find approximating algorithms that overcome the $O(n^2)$ complexity, permitting very large-scale modeling on much less computing devices.

Third, integrating BrainPy models with modern accelerators and neuromorphic computing systems (*Schuman et al., 2017*) could offer a more efficient and scalable approach for simulating large-scale brain dynamics models on cutting-edge hardware accelerators. On the one hand, the implementation of sparse and event-driven operators is necessary for TPUs. While TPUs have demonstrated promising performance and efficiency for machine-learning workloads, our experiments indicate that they are less efficient than GPUs when simulating sparse biological brain network models (see *Appendix 11— figure 8*). This inefficiency is primarily due to the lack of dedicated operators for sparse and event-driven brain computations in current TPUs. In the future, we plan to explore the development of TPU kernels to enable scalable and efficient brain dynamics programming on TPU hardware accelerators. On the other hand, neuromorphic systems incorporate custom analog circuits that mimic neurobiological architectures and dynamics, resulting in significantly higher power efficiency compared to conventional digital hardware. By mapping BrainPy models onto neuromorphic platforms, simulations can be accelerated, and large-scale models can be executed efficiently. However, the development of translation tools and mapping optimizations is necessary to fully harness the potential of these systems.

By addressing these limitations and enhancing BrainPy's capabilities in these areas, we can further advance its goal of serving as a comprehensive programming framework for modeling brain dynamics. This will enable users to delve into the dynamics of brain or brain-inspired models that combine biological insights with machine learning. The BrainPy team encourages collaboration with the research community to expand this modeling ecosystem and facilitate a deeper understanding of brain dynamics.

## Acknowledgements

This work was supported by Science and Technology Innovation 2030-Brain Science and Brain-inspired Intelligence Project (No. 2021ZD0200204) and Beijing Academy of Artificial Intelligence. We would like to express our sincere gratitude to Marcel Stimberg for his valuable insights and assistance with benchmarking brain simulators, which greatly contributed to this paper. We would like to acknowledge Xiaohan Lin, Yifeng Gong, Hongyaoxing Gu, Linfei Lu, Xiaolong Zou, Zhiyu Zhao, Yingqian Jiang, Xinyu Liu, and all other members of the Wu laboratory for their helpful discussions. We thank all GitHub users who contributed codes to BrainPy.

## Additional information

### Funding

| Funder | Grant reference number | Author |
|---|---|---|
| Ministry of Science & Technology, People Republic of China | 2021ZD0200204 | Si Wu |
| Peking-Tsinghua Center for Life Sciences | | Si Wu |

The funders had no role in study design, data collection, and interpretation, or the decision to submit the work for publication.

### Author contributions

Chaoming Wang, Conceptualization, Resources, Software, Formal analysis, Validation, Investigation, Visualization, Methodology, Writing - original draft, Writing - review and editing; Tianqiu Zhang, Xiaoyu Chen, Software, Visualization, Methodology; Sichao He, Software; Shangyang Li, Validation; Si Wu, Conceptualization, Resources, Supervision, Funding acquisition, Writing - original draft, Writing - review and editing

### Author ORCIDs

Chaoming Wang http://orcid.org/0000-0002-7986-3890
Si Wu http://orcid.org/0000-0001-9650-6935

### Decision letter and Author response

Decision letter https://doi.org/10.7554/eLife.86365.sa1
Author response https://doi.org/10.7554/eLife.86365.sa2

## Additional files

### Supplementary files

• MDAR checklist

### Data availability

BrainPy is distributed via the pypi package index (https://pypi.org/project/brainpy/) and is publicly released on GitHub (https://github.com/brainpy/BrainPy/; *Wang et al., 2024*) under the license of GNU General Public License v3.0. Its documentation is hosted on the free documentation hosting platform Read the Docs (https://brainpy.readthedocs.io/). Rich examples and illustrations of BrainPy

are publicly available at the website of https://brainpy-examples.readthedocs.io/. The source codes of these examples are available at https://github.com/brainpy/examples/ (*Wang, 2023*). All the codes to reproduce the results in the paper can be found at the following GitHub repository: https://github.com/brainpy/brainpy-elife-reproducibility/ (copy archived at *Wang, 2024*).

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

## Appendix 1

### Review of the existing programming paradigm

In general, the existing tools for brain dynamics programming can be roughly divided into two categories: low-level programming and descriptive language.

The representatives of the first category include NEURON (*Hines and Carnevale, 1997*), NEST (*Gewaltig and Diesmann, 2007*), CARLsim (*Beyeler et al., 2015*; *Chou et al., 2018*), NeuronGPU (*Golosio et al., 2021*), Arbor (*Akar et al., 2019*), and others. These simulators offer a library of standard models written in C/C++ (particularly for NEST), CUDA (for CARLsim, NeuronGPU, and Arbor), or domain-specific languages (for NEURON and Arbor) to ensure efficient execution, along with a user-friendly Python interface for ease of use. Users can create neural networks in Python by utilizing the neuron and synapse models provided in the library. However, when a new model is required, users must learn to program using the low-level language. This significantly increases the learning cost and restricts the flexibility in defining new models (*Tikidji-Hamburyan et al., 2017*).

The second category tools include Brian (*Goodman and Brette, 2008*), Brian2 (*Stimberg et al., 2014*; *Stimberg et al., 2019*), ANNarchy (*Vitay et al., 2015*), GeNN (*Yavuz et al., 2016*), BMTK (*Dai et al., 2020a*), NetPyNE (*Dura-Bernal et al., 2019*), NeuroML (*Gleeson et al., 2010*), and NMODL (*Krzhizhanovskaya et al., 2020*), which employ a code generation approach based on descriptive languages. Descriptive simulators allow users to create new models based on convenient descriptions (such as text, *Goodman and Brette, 2008*; *Stimberg et al., 2014*; *Stimberg et al., 2019*; *Vitay et al., 2015*; JSON, *Dai et al., 2020a*; *Dura-Bernal et al., 2019*; XML files, *Gleeson et al., 2010*; or customized languages, *Krzhizhanovskaya et al., 2020*) and then translate the descriptions into low-level codes to speed up model running. In such a way, descriptive simulators enable model customization based on high-level descriptive languages and ensure efficient running by generating low-level codes.

Currently, descriptive language has become a standard approach for brain simulation (*Blundell et al., 2018*). Simulators employing this first approach also start to provide their descriptive language interface for code generation. For instance, the NEST simulator provided its domain-specific language NESTML (*Plotnikov et al., 2016*) to describe stereotypical neuron and synapse models. Similarly, the NEURON simulator recently released its modern descriptive interface NetPyNE (*Dura-Bernal et al., 2019*), which employs the standard JSON format to allow users to describe neural circuit models by composing the existing available NEURON building block models. However, users still need to code based on its low-level programming interface to customize new models for channels, neurons, or synapses.

Descriptive languages have been highly successful for brain simulation (*Blundell et al., 2018*). A major benefit of these declarative approaches is the clear separation between model specification and implementation. This frees users from low-level programming details, enabling them to intuitively specify complex models in a concise, semantically clear way. The declarative nature of descriptive languages allows modelers to focus on computational neuroscience instead of implementation specifics. This has enabled rapid prototyping and sharing of models in the field. Overall, descriptive simulation languages have greatly improved modeler productivity thanks to their high-level, implementation-agnostic nature. However, they have intrinsic limitations on transparency, extensibility, and flexibility. One prominent feature of these descriptive languages is that they completely separate the model definition from the simulation, and therefore are not directly executable (*Blundell et al., 2018*). This kind of programming paradigm will cause great restrictions on usability and flexibility because it disables model debugging, error correction, and direct logic controlling. Moreover, descriptive languages are usually designed for specific kinds of models or one particular modeling approach. They are written in more than two programming languages: one is based on a low-level language (e.g., C++, CUDA) to implement its core functionality, and the other is based on a high-level language (e.g., Python, Matlab) for ease of use. Once they are not tailored to users' needs, extensions to accommodate new components must be made in both high- and low-level languages, which is hard or nearly impossible for normal users. What is more, descriptive languages greatly reduce the expressive power of a general-purpose programming language and make it hard to describe all aspects of a simulation experiment, including clipping variables out

of bounds, input–output relations, model debugging, code optimization, dynamics analysis, and others.

In summary, significant challenges to transparency, flexibility, efficiency, and extensibility are still present in the existing programming paradigm for brain simulation. We can draw the conclusion that current software solutions cannot lead us to a general-purpose programming framework that allows us to freely define brain dynamics models in various application domains.

## Appendix 2

### JIT compilation and JIT compilers

A notorious challenge in scientific computing is the trade-off between usability and efficiency. The former seeks the fast prototyping of thoughts and ideas, whereas the latter pursues efficient code execution. For a long time in the past, it was difficult to strike a balance between the two. For example, statically typed compiled programming languages such as C or C++ are incredibly efficient in code execution, but their productivity is relatively low due to their complex and heavy syntax. In contrast, dynamically typed interpreted programming languages like Python and R are easy to learn and use, but they have slow running speeds. Nowadays, with the increasing complexity of models, the demand for both usability and efficiency has increased dramatically. Fortunately, recent advancements in just-in-time (JIT) compilation technology (*Lattner and Adve, 2004*; *Lattner et al., 2020*) have provided viable answers to this two-language problem. In particular, a new generation of computational engines based on JIT compilation (*Aycock, 2003*) has begun to have an impact on a variety of scientific computing disciplines.

The JIT compilation can be seen as the combination of the statically typed compilation and dynamically typed interpretation. It benefits from both the convenience of dynamic high-level languages like Python and the efficiency of static low-level languages such as C++. At the start of a program execution, a JIT compiler acts like an interpreter. It runs your code step by step, and can output the intermediate results at the run-time for debugging. However, if some hot code snippets that are executed frequently, for example, certain functions or loop bodies, are detected or manually labeled, they will be submitted to the JIT compiler for compilation and storage. In this sense, it acts like a statically typed compiler. Once the compiled code snippets are entered again, the program will directly execute the compiler-generated low-level code without time-consuming interpreting again. Hot code snippets can be automatically detected by the JIT compiler, or be manually labeled by users.

JIT compilation is a mature and well-established technology. It has been adopted in modern programming languages like JAVA, Julia, and Python. JAVA language provides JIT compilation in its JAVA virtual machine (JVM) to accelerate the execution of JAVA code (*Grcevski et al., 2004*). Java source code is first compiled into the platform-independent Java bytecode (.class file). Then, JVM loads the bytecode, interprets it, and executes it. To increase the running speed, JVM detects code that is frequently called through the hotspot detection and submits its bytecode to the JIT compiler to compile them into machine code. For the code with lower frequency, executing it through the JVM interpretation can save the time of the JIT compilation; while for the hot code frequently called, JIT compilation can significantly improve the running speed after the code is compiled. However, compared with Python, JAVA has poor ecosystem support for numerical computing. Its JIT is not specialized in numerical computing, but in general domains.

Julia (*Bezanson et al., 2017*), another dynamic high-level programming language, is recently proposed for high-performance scientific computing. Julia features intuitive, productive, and general-purpose syntax inspired by the success of Python, Matlab, and C++. Moreover, it achieves attractive performance through the JIT compilation based on the LLVM compiler infrastructure (*Lattner and Adve, 2004*). In a remarkably short time, Julia has provided excellent routines for mathematical functions, machine-learning algorithms, data processing tools, visualization utilities, and others. However, Julia is still young. Costs, such as lack of familiarity, rough edges, correctness bugs, and continual language changes, are still imposed on normal users.

Python is a well-known and popular interactive dynamic programming language. It has a long history in numerical computing (*Dubois et al., 1996*; *Harris et al., 2020*). The ecosystem of scientific computing, including array programming (*Harris et al., 2020*), scientific algorithms (*Virtanen et al., 2020*), machine learning (*Pedregosa et al., 2011*), deep learning (*Abadi et al., 2016*; *Paszke et al., 2019*; *Frostig et al., 2018*), image processing (*van der Walt et al., 2014*), data analysis and statistics (*McKinney, 2011*), network analysis (*Hagberg and Swart, 2008*), visualization (*Hunter, 2007*), and many others, has been well established in Python. Before Julia, the JIT compilation was introduced into Python by PyPy in 2007. Later, other attempts, including Pyston, Pyjion, Psyco, JitPy, HOPE, etc., are proposed. With a long history of JIT development, Python nowadays has provided mature platforms of JIT compilation focusing on numerical computing. These numerical JIT platforms include Numba (*Lam et al., 2015*), JAX (*Frostig et al., 2018*), and XLA (*Sabne, 2020*). Each of them has its own characteristics.

JAX (*Frostig et al., 2018*) is a flourishing machine-learning library developed by Google. It aims to provide high-level numerical functions to help users fast prototype machine learning ideas. Moreover, these numerical functions can benefit from powerful functional transformations, like automatic differentiation `grad`, JIT compilation `jit`, automatic vectorization `vmap`, and parallelization `pmap`. JAX makes heavy use of XLA (*Sabne, 2020*) (see the following text) for code optimization. Specifically, for ease of use, high-level numerical functions in JAX are NumPy like. JAX provides many numerical functions in NumPy, including basic mathematical operators, linear algebra functions, and Fourier transform routines. However, some fundamental designs are significantly different from NumPy, for instance, the well-established syntax for in-place updating and random samplings. This is the reason why we provide another set of numerical functions consistent with NumPy. In addition to its NumPy-like API, JAX provides a wonderful set of composable functional transformations. Among them, automatic differentiation in JAX supports both forward and backward modes for arbitrary numerical functions. It can take derivatives of a function with a large subset of Python syntax, including loops, conditions, recursions, and closures. Moreover, JAX utilizes XLA to JIT compile your Python code on modern devices, like Central Processing Units (CPUs), Graphics Processing Units (GPUs), and Tensor Processing Units (TPUs). It can significantly accelerate the execution speed of your code and allows you to get maximal performance without having to leave Python. JAX also provides automatic vectorization or batching. It supports transforming loops to vector operations via a single functional call vmap. What's more, JAX delivers pmap to express single-instruction multiple-data programs. Applying pmap to a function will JIT compile and execute the code in parallel on XLA devices, like multiple GPUs or TPU cores. Similar to vmap, pmap transformation maps a function over array axes. But what is different is that the former vectorizes functions by compiling the mapped axis as primitive operations, whereas the latter replicates the function and runs each replica on its own XLA device in parallel. Automatic differentiation and compilation in JAX can be composed arbitrarily to enable rapid experimentation of novel algorithms.

XLA (*Sabne, 2020*) is a domain-specific linear algebra compiler developed by Google that aims to improve the execution speed, memory usage, portability, and mobile footprint reduction of machine-learning algorithms. XLA compiler provides support for JIT compilation based on LLVM (*Lattner and Adve, 2004*). The front-end program (e.g., JAX) which wants to take advantage of the JIT compilation of XLA should first define the computation graph as 'High-Level Optimizer IR' (HLO IR). Then, XLA takes this graph defined in HLO IR and compiles it into machine instructions for different backend architectures. Currently, XLA supports JIT compilation on backend devices of x86-64 CPUs, NVIDIA GPUs, and Google TPUs. XLA is designed for easy portability on new hardware. It provides an abstract interface that a new hardware device can implement to create a backend to run existing computation graphs. Instead of implementing every existing operator for new hardware, XLA provides a simple and scalable mechanism that can retarget different backends. This advantage may be valuable for neuromorphic computing (*Schuman et al., 2017*), because new neuromorphic hardware can be interfaced as a new backend of XLA computation.

Numba (*Lam et al., 2015*) is a JIT compiler for numerical functions in Python. Similar to JAX, Numba supports the JIT compilation of a subset of Python and NumPy code based on the LLVM compiler. However, different from JAX which accelerates the computation flow composed of high-level operators, Numba pays more attention to the acceleration of loop-based functions. Numba achieves excellent optimizations on Python functions with a lot of loops. It allows users to write fine-grained code with native Python control flows and meanwhile obtains the running speed approaching C. This is a huge advantage compared to JAX because JAX does not support the automatic vectorization of a for-loop function.

## Appendix 3

### Continuous integration and documentation generation

To ensure any code changes do not introduce unintended bugs or side effects, we standardized the development process and enabled the automatic continuous integration (CI) of BrainPy with GitHub Actions. Moreover, to update the tutorial and documentation with the latest code changes, we automated the documentation building of BrainPy with *Read the Docs*. *Appendix 3—figure 1* illustrates the whole workflow of BrainPy development. First, any code change should be proposed through GitHub Pull Request. Once a Pull Request is opened, CI pipelines are triggered to test BrainPy codes on Windows, Linux, and macOS platforms (*Appendix 3—figure 1* (1)). After all test suites are passed, the code reviewer should manually inspect the significance and correctness of the proposed code changes again. If all things are fine, the reviewer can merge the code into the master branch (*Appendix 3—figure 1* (2)). After merging, a new set of test cases is triggered automatically to test that the latest BrainPy codebase does not have bugs (*Appendix 3—figure 1* (3)). Besides, the merging operation also triggers the automatic documentation generation through the documentation hosting platform *Read the Docs* (*Appendix 3—figure 1* (4)), in which the latest documentation, including the code annotation, user manual, and tutorials, are automatically built with Sphinx and hosted online at https://brainpy.readthedocs.io/.

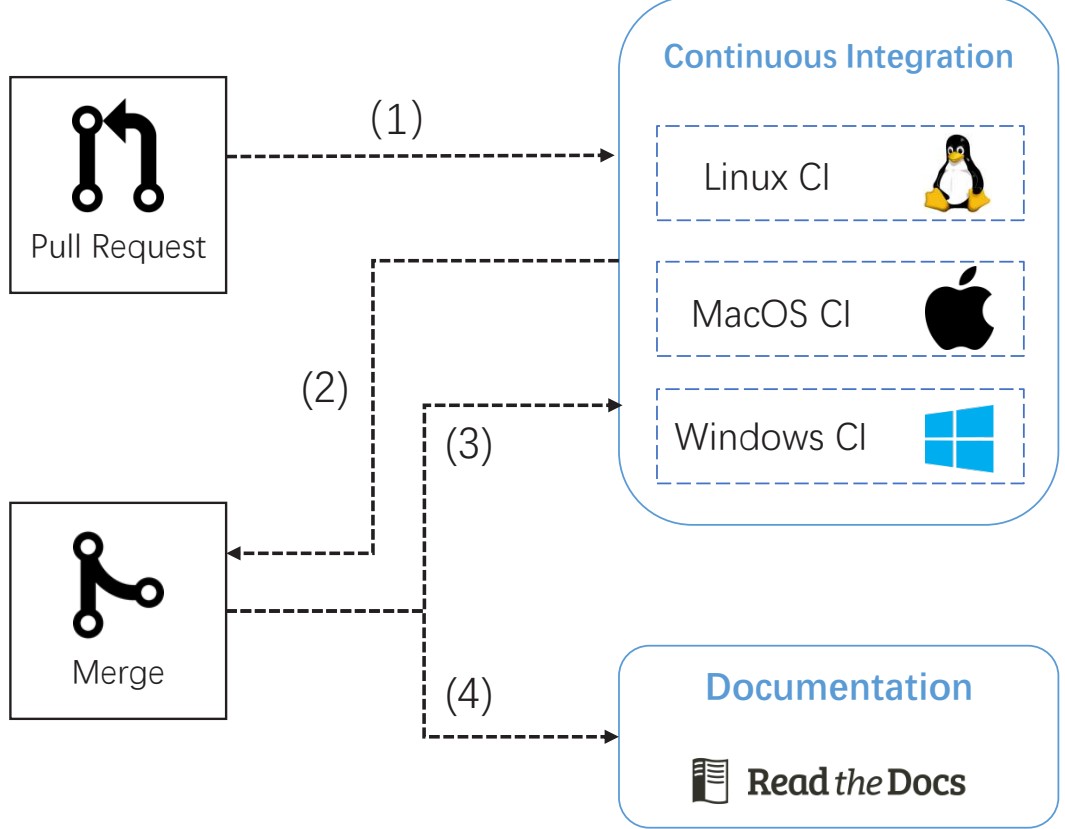

**Appendix 3—figure 1.** The pipeline of automatic continuous integration and documentation building in BrainPy.

## Appendix 4

## Mathematical operators for brain dynamics modeling

Brain dynamics modeling involves conventional computation based on dense matrices and event-driven computation based on sparse connections. BrainPy provides operators for these two kinds of computations. For the list of the number of currently implemented operators please see *Appendix 4—table 1*.

**Appendix 4—table 1.** Number of mathematical operators implemented in BrainPy.
This list will continue to expand since BrainPy will continue to add more operators for brain dynamics modeling. The list of implemented operators is online available at https://brainpy. readthedocs.io/en/latest/apis/math.html.

|  | Number |
|---|---|
| Dense operators with NumPy syntax | 472 |
| Dense operators with TensorFlow syntax | 25 |
| Dense operators with PyTorch syntax | 10 |
| Sparse and event-driven operators | 20 |

## Dense matrix operators

JAX (*Frostig et al., 2018*) has provided most numerical functions in NumPy (*Harris et al., 2020*). However, there are several significant differences between JAX and NumPy. First, the array structure in JAX does not support in-place updating. Second, numerical functions that need in-place updating are missing in JAX. Third, random sampling functions are significantly different from NumPy.

The dense matrix operators in BrainPy are based on JAX's implementations but are designed to be seamlessly consistent with NumPy. First, we provide `brainpy.math.Array` which is consistent with NumPy's ndarray structure, and a series of mathematical operations for `brainpy.math.Array` which is similar to those for ndarray. Second, mathematical operators for `brainpy.math.Array`, such as indexing, slicing, sorting, rounding, arithmetic operations, linear algebraic functions, and Fourier transform routines, are all supported. Many of these operators (nearly 85%) are directly implemented through the NumPy-like functions in JAX, while BrainPy provides dozens of APIs missing or inconsistent in JAX. Third, to unify random number generators, BrainPy implements most of the random sampling functions in NumPy, including its univariate distributions, multivariate distributions, standard distributions, and utility functions.

Moreover, BrainPy is working on dense operators provided in PyTorch (*Paszke et al., 2019*) and TensorFlow (*Abadi et al., 2016*) libraries. In the future, BrainPy will continue to cover dense array operators in TensorFlow and PyTorch, since these implementation syntaxes have been widely accepted in the community.

## Dedicated operators

Brain dynamics models differ from deep neural network (DNN) models in the way they perform computation. Brain dynamics models typically have sparse connections (less than 20% of neurons are connected to each other) and perform event-driven computations (synaptic currents are only transmitted when a presynaptic neuron spikes). These unique features make brain dynamics models less efficient when conventional dense array operators are used. To tackle this efficiency issue, traditional brain simulators heavily rely on event-driven synaptic operations. Previous works have explored event-driven synaptic operations on both CPU platforms (see *Vitay et al., 2015*; *Plesser et al., 2007*; *Stimberg et al., 2019*) and GPU platforms (see *Fidjeland et al., 2009*; *Brette and Goodman, 2011*; *Yavuz et al., 2016*; *Alevi et al., 2022*).

Despite the effectiveness of these simulators, one limitation is the lack of abstraction of event-driven synaptic operations as primitive low-level operators. In other words, these operations are not treated as fundamental building blocks that can be easily manipulated and optimized. This absence of abstraction hampers the development of more efficient algorithms and restricts the flexibility and extensibility of the simulators. Therefore, it is crucial to bridge this gap and provide a higher level of abstraction for event-driven synaptic operations, which we refer to as '*event-driven operators*'. Note here in BrainPy, event-driven operators are employed within a *clock-driven simulation* schema, where

the simulation advances in a synchronized manner, updating all neurons and synapses at each time step. These event-driven operators process information based on the detection of spatial spiking events. They execute computations when the presynaptic neuron fires spikes at each time step. This contrasts with an *event-driven simulation* approach (*Ros et al., 2006*), where neuronal or synaptic state updates are triggered by temporal spiking events, rather than updating all elements at each time step. Therefore, the key distinction between our event-driven operators and an event-driven simulation scheme lies in their scope and application: event-driven operators serve as fundamental building blocks that define how individual components of an SNN respond to spatial events at the current time step, while an event-driven simulation scheme serves as a methodology for simulating the collective behavior of the entire network based on the occurrence of temporal spikes.

Moreover, by abstracting these operations into primitive low-level operators, BrainPy offers automatic differentiation support compatible with `jax.grad`. It also enables vectorization and parallelization support compatible with `jax.vmap` and `jax.pmap`. This compatibility further enhances the applicability of event-driven operators across a wider range of contexts.

Specifically, BrainPy provides these sparse and event-driven operators in the following modules: (1) The `brainpy.math.sparse` module. This module provides a set of sparse operators that can store synaptic connectivity compactly and compute synaptic currents efficiently. (2) The `brainpy.math.event` module. This module implements event-driven operators that only perform computations when spikes arrive. This can lead to significant improvements in efficiency, as the state of the system does not need to be constantly updated when no spikes arrive.

In addition, efficient modeling of brain dynamics encounters scalability issues. The computational resources and device memory requirements for brain dynamics models increase quadratically with the number of neurons, as the synaptic connectivity grows almost quadratically in relation to the number of neurons. This characteristic severely restricts the size of the network that can be simulated on a single device using traditional array programming solutions.

One established approach to address this challenge is the utilization of JIT connectivity (*Lytton et al., 2008*; *Azevedo Carvalho et al., 2020*; *Knight and Nowotny, 2021*). This method involves regenerating the synaptic connectivity during computation by controlling the same random seed. Specifically, the `brainpy.math.jitconn` module provides JIT connectivity as primitive operators, specifically designed for cases where synaptic connections follow a fixed connectivity rule and do not require modifications after initialization. These operators eliminate the memory overhead required for connectivity storage, as synaptic connectivity can be generated JIT during the execution of the operators. Notably, when compared to conventional operators, they enable the execution of large-scale networks that are two to three orders of magnitude larger on a single device.

Moreover, BrainPy also provides operators that are involved in transformations among presynaptic neuronal data, synaptic data, and postsynaptic neuronal data in the brainpy.math module. Specifically, this module provides operators for mapping presynaptic neuronal data to its connected synaptic dimension (pre-to-syn), arithmetic operators including summation, product, max, min, mean, and softmax for transforming synaptic data to postsynaptic neuronal dimension (syn-to-post), and arithmetic operators such as summation, product, max, min, and mean for directly computing postsynaptic data from presynaptic events (pre-to-post).

## Appendix 5

### Numerical solvers for differential equations

To meet the large demand for solving differential equations in brain dynamics modeling, BrainPy implements numerical solvers for ODEs, SDEs, fractional differential equations (FDEs), and delay differential equations (DDEs). In general, numerical integration in BrainPy defines the system evolution of $x(t) \rightarrow x(t + dt)$, where $x$ is the state, $t$ is the current time, and $dt$ is the integration step.

### ODE numerical solvers

In BrainPy, the integration of an ODE system $dx/dt = f(x, t)$ is performed as

```
1 integral=brainpy.odeint(f=<function>,
2                         method=<str>,
3                         dt=<float>)
```

where method denotes the numerical method used to integrate the ODE function, and dt controls the initial or default numerical integration step. A variety of numerical integration methods for ODEs, including Runge–Kutta, adaptive Runge–Kutta, and Exponential methods, are supported in BrainPy (see *Appendix 5—table 1*).

**Appendix 5—table 1.** Numerical solvers provided in BrainPy for ordinary differential equations.

| Solver type | Solver name | Keyword |
|---|---|---|
| | Euler | euler |
| | Midpoint | midpoint |
| | Heun's second-order method | heun2 |
| | Ralston's second-order method | ralston2 |
| | Second-order Runge–Kutta method | rk2 |
| | Third-order Runge–Kutta method | rk3 |
| | Four-order Runge–Kutta method | rk4 |
| | Heun's third-order method | heun3 |
| | Ralston's third-order method | ralston3 |
| | Third-order strong stability preserving Runge–Kutta method | ssprk3 |
| | Ralston's fourth-order method | ralston4 |
| Runge–Kutta method | Fourth-order Runge–Kutta method with 3/8-rule | rk4_38rule |
| | Runge–Kutta–Fehlberg 4 (5) | rkf45 |
| | Runge–Kutta–Fehlberg 1 (2) | rkf12 |
| | Dormand–Prince method | rkdp |
| | Cash–Karp method | ck |
| | Bogacki–Shampine method | bs |
| Adaptive Runge–Kutta method | Heun–Euler method | heun_euler |
| Exponential method | Exponential Euler method | exp_euler |

### SDE numerical solvers

For a general SDE system with multi-dimensional driving Wiener processes,

$$dx = f(x, t, p_1)dt + \sum_{\alpha=1}^{m} g_\alpha(x, t, p_2)dW^\alpha,$$

BrainPy supports its numerical integration with

```
1 integral = brainpy.sdeint(f=<function > ,
2                           g=<function > ,
3                           method =<str > ,
4                           dt =<float > ,
5                           wiener_type=<'scalar' or 'vector'>,
6                           intg_type=<'Ito' or 'Stratonovich'>)
```

where `method` specifies the numerical solver, `dt` the default integral step, `wiener_type` the type of Wiener process (`SCALAR_WIENER` for scalar noise or `VECTOR_WIENER` for multi-dimensional driving Wiener processes), and integral_type the integral type (`ITO_SDE` for the Itô integral or `STRA_SDE` for the Stratonovich stochastic integral). See *Appendix 5—table 2* for the full list of SDE solvers currently implemented in BrainPy.

**Appendix 5—table 2.** Numerical solvers provided in BrainPy for stochastic differential equations. 'Y' denotes support. 'N' denotes not support.

| Integral type | Solver name | Keyword | Scalar Wiener | Vector Wiener |
|---|---|---|---|---|
| | Strong SRK scheme: SRI1W1 | srk1w1_scalar | Y | N |
| | Strong SRK scheme: SRI2W1 | srk2w1_scalar | Y | N |
| | Strong SRK scheme: KlPl | KlPl_scalar | Y | N |
| | Euler method | euler | Y | Y |
| | Milstein method | milstein | Y | Y |
| | Derivative-free Milstein method | milstein2 | Y | Y |
| Itô integral | Exponential Euler | exp_euler | Y | Y |
| | Euler method | euler | Y | Y |
| | Heun method | heun | Y | Y |
| | Milstein method | milstein | Y | Y |
| Stratonovich integral | Derivative-free Milstein method | milstein2 | Y | Y |

## FDE numerical solvers

The numerical integration of FDEs is very similar to that of ODEs, except that the initial value, memory length, and fractional order should be provided. Given the fractional-order differential equation

$$\frac{\mathrm{d}^\alpha x}{\mathrm{d}t^\alpha} = F(x, t),$$

where the fractional order $0 < \alpha \leq 1$. BrainPy supports its numerical integration with the following format of

```
1 brainpy.fdeint(f=<function> ,
2                alpha=<float> ,
3                num_memory=<int> ,
4                inits=<dict> ,
5                method=<str> )
```

BrainPy supports two types of FDE, that is, the Caputo derivative and Grünwald–Letnikov derivative. Caputo derivatives are widely used in neuroscience modeling (***Kaslik and Sivasundaram, 2012***). However, the numerical integration of Caputo derivatives has a high memory consumption, because it requires integration over all past activities. This implies that FDEs with the Caputo derivative definition cannot be used to simulate realistic neural systems. However, the numerical method for Grünwald–Letnikov FDEs, `brainpy.fde.GLShortMemory`, is highly efficient because it does not require an infinite memory length for numerical solutions. With the increasing length of

short memory, the accuracy of the numerical approximation will increase. Therefore, `brainpy.fde.GLShortMemory` can be applied to real-world neural modeling. See *Appendix 5—table 3* for the full list of FDE solvers currently implemented in BrainPy.

**Appendix 5—table 3.** Numerical solvers provided in BrainPy for fractional differential equations.

| Derivative type | Solver name | Keyword |
| --- | --- | --- |
| | L1 schema | l1 |
| Caputo derivative | Euler method | euler |
| Grünwald–Letnikov derivative | Short Memory Principle | short-memory |

## DDE numerical solvers

Delays occur in any type of differential equation. In realistic neural modeling, delays are often inevitable. BrainPy supports equations with variables of constant delays, like

$$y'(t) = f(t, y(t), y(t - \tau_1), y(t - \tau_2), \ldots, y(t - \tau_k)),$$

where the time lags $\tau_j$ are the positive constants. It also supports systems with state-dependent delays, for example,

$$y'(t) = f(t, y(t), y(t - f_1(y(t))), \ldots, y(t - f_k(y(t)))),$$

where $f_k$ is the function that computes the delay length by the system state $y(t)$. For neutral-typed equations in which delays appear in derivative terms,

$$y'(t) = f(t, y(t), y'(t - \tau_1), y'(t - \tau_2), \ldots, y'(t - \tau_k))$$

BrainPy also supports its numerical integration.

BrainPy, in particular, implements interfaces to define these various delay variables. `brainpy.math.TimeDelay` and `brainpy.math.LengthDelay` are provided to support state-dependent variables. Both `brainpy.math.TimeDelay` and `brainpy.math.LengthDelay` are used to delay neuronal signals in BrainPy, for example, Boolean spikes, real-valued synaptic conductance, or membrane potentials.

The `TimeDelay` is the class representing a connection delay measured in time units. For example, one might specify a $T$ ms delay. The TimeDelay class stores history values at times $[T_0 - T, T_0 - T + \Delta t, \cdots, T_0 - 2\Delta t, T_0 - \Delta t, T_0]$, where $T_0$ is the current time and $\Delta t$ is the time step. Users can retrieve the history values for any $t$ in the interval $[T_0 - T, T_0]$. For a $t$ that exactly matches one of the stored time points, TimeDelay directly returns the stored history values; otherwise, TimeDelay uses linear interpolation to estimate values between the nearest stored data points, or rounding to return the value from the nearest data point in time.

The `LengthDelay` class represents a connection delay based on the number or length of running iterations. For instance, one can specify a delay of $L$ iterations. It stores historical values at previous time steps, such as $[L, L - 1, \cdots, 1, 0]$. Unlike `TimeDelay`, users can only retrieve the historical values at specific discrete time steps that have been stored.

Despite the distinct usage characteristics of `TimeDelay` and `LengthDelay`, both of them employ a circular array for their delay updates. This implementation involves utilizing an array of fixed length along with a cursor that indicates the current updating position within the array (see *Appendix 5—figure 1*).

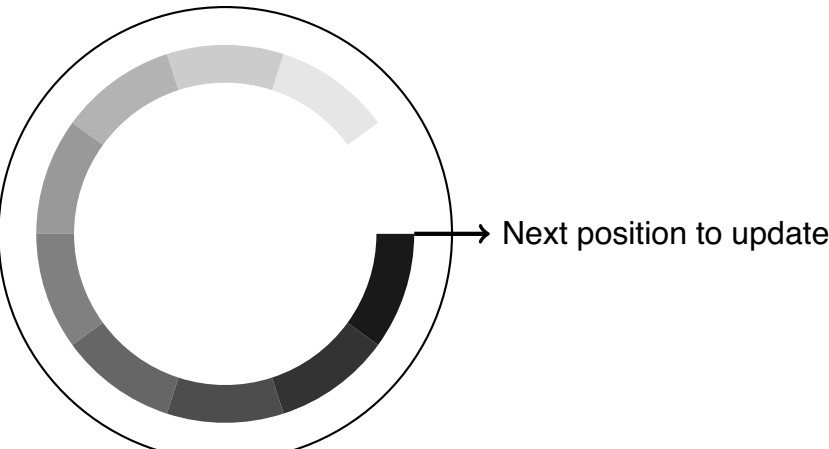

**Appendix 5—figure 1.** The dynamic array to store the delay buffer.

The classes `brainpy.math.NeuTimeDelay` and `brainpy.math.NeuLenDelay` are identical to `brainpy.math.TimeDelay` and `brainpy.math.LengthDelay`, respectively. However, they are specifically designed to model neutral delay variables.

With these delay supports, the differential equations with delayed variables are intrinsically supported in each integral function.

For delayed ODEs, users can use

```
1 brainpy.odeint(f=<function> ,
2                method=<str> ,
3                dt=<float> ,
4                state_delays=<dict> ,
5                neutral_delays=<dict>)
```

Similarly, the numerical integration of delayed SDEs should utilize

```
1 brainpy.sdeint(f=<function> ,
2                g=<function> ,
3                method=<str> ,
4                dt=<float> ,
5                state_delays=<dict>)
```

For delayed FDEs, one can use

```
1 brainpy.fdeint(f=<function> ,
2                method=<str> ,
3                dt=<float> ,
4                num_memory=<int> ,
5                inits=<dict>,
6                state_delays=<dict>)
```

Note that we currently do not support neutral delays for delayed SDEs and FDEs. However, users can easily customize their supports for equations with neutral delays.

## Appendix 6

### Object-oriented JIT compilation and automatic differentiation

Under minimal constraints and assumptions, BrainPy enables the JIT compilation for class objects. These assumptions include the following:

- The class for JIT must be a subclass of brainpy.BrainPyObject.
- Dynamically changed variables in the class must be labeled as brainpy.math.Variable. Otherwise, they will be compiled as constants and cannot be updated during the program execution.

To take advantage of the JIT compilation, we can directly apply brainpy.math.jit() onto the instantiated class objects, or functions of a class object.

`brainpy.math.grad()` takes a function/object ($f: \mathbb{R}^n \to \mathbb{R}$, which returns a scalar value) as the input and returns a new function ($\partial f(x) \to \mathbb{R}^n$) which computes the gradient of the original function/object.

```
1 grad_f=brainpy.math.grad(f, grad_vars=<variables to take gradients>)
```

`brainpy.math.vector_grad()` takes vector-valued gradients for a function/object ($f: \mathbb{R}^n \to \mathbb{R}^m$). It returns a new function ($\partial f(x): \mathbb{R}^m \to \mathbb{R}^n$) which evaluates the vector-Jacobian products.

```
1 grad_f=brainpy.math.vector_grad(f, grad_vars<variables to take
gradients>)
```

Another way to take gradients of a vector-output value is using brainpy.math.jacobian(). It aims to automatically compute the Jacobian matrices $\partial f(x) \in \mathbb{R}^{m \times n}$ by the given function/object $f: \mathbb{R}^n \to \mathbb{R}^m$ at the given point of $x \in \mathbb{R}^n$.

```
1 grad_f=brainpy.math.jacobian(f, grad_vars<variables to take gradients>)
```

## Appendix 7

## Extension of BrainPy infrastructure

BrainPy features extensible architecture. New extensions can be easily made by using BrainPy infrastructure. Even new tools at the infrastructure level can be customized by using BrainPy operators.

At the toolbox level, BrainPy provides a mechanism in which the extension of a new tool can be made by adding a new subclass. For instance, a new synaptic connection method can be extended by subclassing `brainpy.connect.TwoEndConnector`:

```
1 class NewConnector(bp.conn.TwoEndConnector):
2   def __init__(self):
3    # initializing connector
4
5   def build_csr(self):
6    # build the CSR data structure
7
8   def build_coo(self):
9    # build the COO data structure
```

As another example, customizing a new weight initialization method can be added by inheriting from brainpy.initialize.Initializer base class:

```
1 class NewInitializer(bp.init.Initializer):
2   def __init__(self):
3    # initializing connector
4
5   def __call__(self, shape, dtype =None):
6    # building weights
```

At the model building level, BrainPy enables to flexibly customize the user's own dynamical systems by simply subclassing brainpy.DynamicalSystem base class.

At the numerical integrator level, BrainPy provides an easy way to write new integrator methods. For example, adding a new Runge–Kutta integrator can be done by subclassing `brainpy.ode.ExplicitRKIntegrator` and providing the corresponding Butcher tableau:

```
1 class NewExplicitRK(bp.ode.ExplicitRKIntegrator):
2   A = [] # The A matrix in the Butcher tableau.
3   B = [] # The B vector in the Butcher tableau.
4   C = [] # The C vector in the Butcher tableau.
```

In a similar way, a customized adaptive Runge–Kutta integrator can be extended by subclassing `brainpy.ode.AdaptiveRKIntegrator` with the specification of the corresponding Butcher tableau:

```
1 class NewAdaptiveRK(bp.ode.ExplicitRKIntegrator):
2   A = [] # The A matrix in the Butcher tableau.
3   B1 = [] # The B1 vector in the Butcher tableau.
4   B2 = [] # The B2 vector in the Butcher tableau.
5   C = [] # The C vector in the Butcher tableau.
```

At the operator level, BrainPy is trying to provide a way to write fine-granularity low-level operators in the same Python interface by adopting the same JIT compilation technology (please refer to Appendix 8).

## Appendix 8

## Extension of low-level operators in BrainPy

By bridging Numba (*Lam et al., 2015*), JAX (*Frostig et al., 2018*), and XLA (*Sabne, 2020*), BrainPy enables the customization of primitive operators through the native Python syntax. Exposing a custom operator to JAX requires registering an XLA 'custom call', and providing its C callback for Python. Based on the following two properties of Numba, we are aware of the possibility of using Numba as a convenient method for writing low-level kernels that support JAX's JIT compilation. First, unlike JAX, which only supports the JIT compilation of high-level functions, Numba is a JIT compiler that allows users to write a function with low-level fine-grained operations, for instance, looping over an array, or conditional branching over array values. This implies that `numba.jit()` can be used as a means to write low-level kernel functions. The second property of Numba is that it provides a mechanism to create a compiled function that is callable from the foreign C code, such as XLA. Specifically, `numba.cfunc()` can be used to create a C callback for Numba's JIT function to interface with XLA. Therefore, by integrating Numba with JAX and XLA, BrainPy provides an interface where users write primitive operators directly with Python syntax. Note that Numba supports various native Python features and a large subset of NumPy functions. Therefore, there is a large flexibility in coding low-level primitives with Numba.

Below, we illustrate how to write a primitive operator in BrainPy. Particularly, to customize a primitive operator we need to provide two functions. The first is an abstract evaluation function that tells JAX what shapes and types of outputs are produced according to the input shapes and types:

```
1 def abstract_eval(*ins):
2   return outs
```

in which `ins` specifies the information of input shapes and types, outs denotes the array information of shapes and types of outputs. The other function is the concrete computation function, in which the output data is calculated according to the input data:

```
1 def concrete_comp(outs, ins):
2   # calculate outputs according to inputs
```

where outs and ins are output and input data owned by XLA, respectively. Note this function should not return anything. All computations must be made in place. Finally, by using

```
1 # "name" is the operator name
2 # "concrete_comp" is the concrete computation function
3 # "abstract_eval" is the abstract evaluation function
4 op =brainpy.math.CustomOpByNumba(abstract_eval, concrete_comp, name)
```

we register a primitive operator op. This operator op can be used anywhere the user wants. The Numba-based approach for operator customization demonstrates comparable performance to the native C++-based XLA operator registration. To illustrate this, we conducted a performance comparison of the COBA network model (*Vogels and Abbott, 2005*) using both an event-driven operator implementation based on Numba and C++.

This event-driven operator implements the matrix–vector multiplication $\mathbf{y} = \mathbf{v} \cdot \mathbf{M}$ for synaptic computation, where $\mathbf{v}$ is the presynaptic spikes, $\mathbf{M}$ the synaptic connection matrix, and $\mathbf{y}$ the postsynaptic current. Specifically, it performs matrix–vector multiplication in a sparse and efficient way by exploiting the event property of the input vector $\mathbf{v}$. Particularly, we performs event-driven matrix–vector multiplication only for the non-zero elements of the vector, which are called events. This can reduce the number of operations and memory accesses, and improve the running performance of matrix–vector multiplication.

Based on our Python-level registration syntax, this operator can be implemented as:

```
1 from jax.core import ShapedArray
2 import brainpy.math as bm
```

```
3
4 # the abstract evaluation function
5 def abs_eval(events, indices, indptr, *, weight, post_num):
6     return [ShapedArray((post_num,), bm.float32), ]
7
8 # the concrete function implementing the event-driven computation
9 def con_compute(outs, ins):
10    post_val, =outs
11    post_val.fill(0)
12    events, indices, indptr, weight, _=ins
13    weight=weight[()]
14    for i in range(events.size):
15      if events[i]:
16        for j in range(indptr[i], indptr[i+1]):
17          index =indices[j]
18          post_val[index]+=weight
19
20 # operator registration
21 event_sum =bm.CustomOpByNumba(eval_shape =abs_eval, cpu_func =con_compute)
```

Listing 1: The event-driven operator implemented with our Python syntax for the computation of $\mathbf{y} = \mathbf{v} \cdot \mathbf{M}$.

This operator can also be implemented with the pure C++ syntax:

```
1 #include <cstdint>
2 #include <cstring>
3 #include <cmath>
4
5 template <typename F, typename I>
6 void cpu_csr_event_sum_homo(void *out, const void **in) {
7     const std::uint32_t pre_size = *reinterpret_cast<const std::uint32_t *>(in[0]);
8     const std::uint32_t post_size = *reinterpret_cast<const std::uint32_t *>(in[1]);
9     const bool *events=reinterpret_cast<const bool *>(in[2]);
10    const I *indices=reinterpret_cast <const I *>(in[3]);
11    const I *indptr=reinterpret_cast <const I *>(in[4]);
12    const F weight=reinterpret_cast<const F *>(in[5])[0];
13    F *result=reinterpret_cast<F *>(out);
14
15    // algorithm
16    memset(&result[0], 0, sizeof(F) * post_size);
17    for (std::uint32_t i=0; i<pre_size; ++i) {
18      if (events[i]){
19        for (I j=indptr[i]; j<indptr[i+1]; ++j) {
20          result[indices[j]]+=weight;
21        }
22      }
23    }
24 }
```

Listing 2: The event-driven operator utilizes C++ for the computation of $\mathbf{y} = \mathbf{v} \cdot \mathbf{M}$. Here, we present the main code snippet that implements the event-driven matrix–vector multiplication. Please note that we have omitted the code for Python binding, compilation, registration, and other related

aspects. For detailed instructions and a comprehensive tutorial on XLA operator customization, we encourage users to refer to the appropriate resource.

The speed comparison between the two approaches has been depicted in *Figure 9*. Although the approach shows promising results on CPU hardware, it is not directly compatible with other computing platforms like GPUs. This restricts the applicability and scalability of the proposed method, as GPUs are increasingly used to accelerate brain dynamics models. To overcome this limitation, currently, we are working on alternative approach that can be used in both CPU and GPU devices, allowing for broader utilization of available hardware resources and unlocking new possibilities for customizing any complex brain dynamics operators.

## Appendix 9

## Multi-scale spiking neural network for the visual system modeling

We build a spiking network model examined in *Figure 2* and *Figure 3* for modeling the visual system (V1, V2, V4, TEO, and TEpd). Simulations are performed using a network of Hodgkin–Huxley neurons, with the local circuit and long-range connectivity structure derived from *Markov et al., 2014*. Each of the five areas consists of 2000 neurons, with 1600 excitatory and 400 inhibitory neurons.

For each neuron, the membrane potential dynamics is modified from *Traub and Miles, 1991* and is described by the following equations:

$$C_m \frac{dV}{dt} = -g_L \left(V - E_L\right) - \bar{g}_{Na} m^3 h \left(V - E_{Na}\right) - \bar{g}_{Kd} n^4 \left(V - E_K\right) + G(t) \tag{1}$$

where $V$ is the membrane potential, $G(t)$ stands for synaptic interactions, $C_m$ is the membrane capacitance per unit area, $E_K = -90\,\text{mV}$ and $E_{Na} = 50\,\text{mV}$ are the potassium and sodium reversal potentials, respectively, $E_l = -60\,\text{mV}$ is the leak reversal potential, $\bar{g}_K = 30\,\text{mS/cm}^2$ and $\bar{g}_{Na} = 100\,\text{mS/cm}^2$ are the potassium and sodium conductance per unit area, respectively, and $\bar{g}_l = 0.05\,\text{mS/cm}^2$ is the leak conductance per unit area.

Each neuron is composed of two active ion channels, including the potassium and sodium channels. Because the potassium and sodium channels are voltage sensitive, according to the biological experiments, $m$, $n$, and $h$ are used to simulate the activation of the channels.

$$\frac{dm}{dt} = \alpha_m(V)(1 - m) - \beta_m(V)m \tag{2}$$

$$\frac{dh}{dt} = \alpha_h(V)(1 - h) - \beta_h(V)h \tag{3}$$

$$\frac{dn}{dt} = \alpha_n(V)(1 - n) - \beta_n(V)n \tag{4}$$

Specifically, $n$ measures the activation of potassium channels, and $m$ and $h$ measures the activation and inactivation of sodium channels, respectively. $\alpha_x$ and $\beta_x$ are rate constants for the ion channel $x$ and depend exclusively on the membrane potential. The voltage-dependent expressions of the rate constants were modified from the model described by *Traub and Miles, 1991*:

$$\alpha_m = 0.32 * \left(13 - V + V_T\right) / \left[\exp\left(\left(13 - V + V_T\right)/4\right) - 1\right] \tag{5}$$

$$\beta_m = 0.28 * \left(V - V_T - 40\right) / \left[\exp\left(\left(V - V_T - 40\right)/5\right) - 1\right] \tag{6}$$

$$\alpha_h = 0.128 * \exp\left(\left(17 - V + V_T\right)/18\right) \tag{7}$$

$$\beta_h = 4/ \left[1 + \exp\left(\left(40 - V + V_T\right)/5\right)\right] \tag{8}$$

$$\alpha_n = 0.032 * \left(15 - V + V_T\right) / \left[\exp\left(\left(15 - V + V_T\right)/5\right) - 1\right] \tag{9}$$

$$\beta_n = 0.5 * \exp\left(\left(10 - V + V_T\right)/40\right), \tag{10}$$

where $V_T = -63\,\text{mV}$ adjusts the threshold.

For the synapse, we use conductance-based synaptic interactions. Particularly, $G(t)$ is given by:

$$G(t) = -\sum_j g_{ji}(t) \left(V_i - E_j\right), \tag{11}$$

where $V_i$ is the membrane potential of neuron $i$. The synaptic conductance from neuron $j$ to neuron $i$ is represented by $g_{ji}(t)$, while $E_j$ signifies the reversal potential of that synapse. For excitatory synapses, $E_j$ was set to 0 mV, whereas for inhibitory synapses, it was −80 mV. The dynamics of the synaptic conductance is given by:

$$\frac{dg_{ji}}{dt} = -\frac{g_{ji}}{\tau_{\text{decay}}} + g_{\text{max}} \sum_k \delta(t - t_j^k), \tag{12}$$

where $t_j^k$ is the spiking time of the presynaptic spike. Whenever a spike occurred in neuron $j$, the synaptic conductance $g_{ji}$ experienced an immediate increase by a fixed amount $g_{\text{max}}$. Subsequently, the conductance $g_{ji}$ decayed exponentially with a time constant of $\tau_{\text{decay}} = 5$ ms for excitation and $\tau_{\text{decay}} = 10$ ms for inhibition.

The connection density is set according to the experimental connectivity data (*Markov et al., 2014*). The inter-areal connectivity is measured as a weight index (see *Appendix 9—table 1*), called the extrinsic fraction of labeled neurons (*Markov et al., 2014*). The intra-area connectivity is set according to the setting in a standard E/I balanced network (*Brette et al., 2007*).

**Appendix 9—table 1.** The weighted connectivity matrix among five brain areas: V1, V2, V4, TEO, and TEpd (*Markov et al., 2014*).

|      | V1     | V2       | V4     | TEO      | TEpd     |
|------|--------|----------|--------|----------|----------|
| V1   | 0.0    | 0.7322   | 0.1277 | 0.2703   | 0.003631 |
| V2   | 0.7636 | 0.0      | 0.1513 | 0.003274 | 0.001053 |
| V4   | 0.0131 | 0.3908   | 0.0    | 0.2378   | 0.07488  |
| TEO  | 0.0    | 0.02462  | 0.2559 | 0.0      | 0.2313   |
| TEpd | 0.0    | 0.000175 | 0.0274 | 0.1376   | 0.0      |

Moreover, we introduce distance-dependent inter-areal synaptic delays by assuming a conduction velocity of 3.5 m/s (*Swadlow, 1990*) and using a distance matrix based on experimentally measured wiring distances across areas (*Markov et al., 2014*).

**Appendix 9—table 2.** The delay matrix (ms) among five brain areas: V1, V2, V4, TEO, and TEpd (*Markov et al., 2014*).

|      | V1     | V2     | V4     | TEO    | TEpd   |
|------|--------|--------|--------|--------|--------|
| V1   | 0.     | 2.6570 | 4.2285 | 4.2571 | 7.2857 |
| V2   | 2.6570 | 0.     | 2.6857 | 3.4857 | 5.6571 |
| V4   | 4.2285 | 2.6857 | 0.     | 2.8    | 4.7143 |
| TEO  | 4.2571 | 3.4857 | 2.8    | 0.     | 3.0571 |
| TEpd | 7.2857 | 5.6571 | 4.7143 | 3.0571 | 0.     |

## Appendix 10

### Training reservoir computing model with multiple algorithms

Reservoir computing is a type of recurrent neural network that is often used for processing temporal data. Unlike traditional recurrent neural networks, reservoir computing fixes the weights of the recurrent layer (known as the 'reservoir') and only trains the weights of the output layer. This makes training much more efficient.

A reservoir computing model can be trained using various algorithms, such as online learning, offline learning, and backpropagation learning, to optimize its performance. Online learning refers to the process of updating the model in real time as new data becomes available. This algorithm allows the model to adapt and adjust its predictions continuously. Offline learning, on the other hand, involves training the model using a fixed dataset, where the entire dataset is used to update the model's parameters. This method is particularly useful when a large amount of data is available upfront. Lastly, backpropagation learning utilizes a technique called backpropagation to train the model by computing the gradients of the model's parameters and adjusting them accordingly.

The unified building and training interface of BrainPy enables the training of the same reservoir model with multiple training algorithms. By employing BrainPy, we can compare the performance of different training algorithms and determine which approach yields the best results for the reservoir computing model.

The following lists the details of such training.

### Reservoir model

The dynamics of the reservoir model used here are given by:

$$\mathbf{x}(t) = (1 - \alpha) \cdot \mathbf{x}(t - 1) + \alpha \cdot f(\mathbf{W}_{in}\,\mathbf{u}(t) + \mathbf{W}_{rec}\,\mathbf{x}(t - 1)), \tag{13}$$

$$\mathbf{y}(t) = \mathbf{W}_{out}\,\mathbf{x}(t) \tag{14}$$

where $\mathbf{x}(t)$ is the reservoir state, $\mathbf{y}(t)$ the readout value, $\mathbf{W}_{in}$ and $\mathbf{W}_{rec}$ are input and recurrent connection matrices, respectively, $\mathbf{W}_{out}$ the readout weight matrix which can be trained by either offline learning or online learning algorithms, $\alpha \in (0, 1]$ the leaky rate, $\mathbf{u}(t)$ the input at time step $t$, and $f$ the nonlinear activation function.

In BrainPy, the reservoir model can be easily instantiated as the following code:

```
1 reservoir =brainpy.dyn.Reservoir(input_shape, output_shape, leaky_rate)
```

### Inferring Lorenz strange attractor

The reservoir model is utilized for inference tasks in this work. To generate training and testing data, we numerically integrate a simplified model of a weather system originally developed by *Lorenz, 1963*. This model comprises three coupled nonlinear differential equations:

$$\dot{x} = 10(y - x), \tag{15}$$

$$\dot{y} = x(28 - z) - y, \tag{16}$$

$$\dot{z} = xy - 8z/3. \tag{17}$$

The state $X(t) = [x(t), y(t), z(t)]^T$ represents a vector of Rayleigh–Bénard convection observables. The system exhibits deterministic chaos, displaying sensitivity to initial conditions and forming a strange attractor in the phase space trajectory (*Figure 4D*).

In this task, we train a reservoir model to predict the $T$-step-ahead value of all Lorenz variables, $x$, $y$, and $z$. During training, we provide the model with all three variables. During testing, we only provide the model with $x$, $y$, and $z$ at $T$-step-ahead, and it must infer $x, y$, and $z$ after $T$ steps. Here, we choose $T = 5$.

In this task, the input size was set to 3, the recurrent layer of the reservoir model contained 100 units, and the output size was 3, as used in *Figure 4D-F*.

## Training with ridge regression

The training objective of reservoir models is to find the optimal $\mathbf{W}_{out}$ that minimizes the square error between $\mathbf{y}(t)$ and $\mathbf{y}^{target}(t)$. The common way to learn the linear output weight $\mathbf{W}_{out}$ is using the ridge regression algorithm. The ridge regression, also known as regression with Tikhonov regularization, is given by:

$$\mathbf{W}^{out} = \mathbf{Y}^{target} \mathbf{X}^{T} \left( \mathbf{X}\mathbf{X}^{T} + \beta \mathbf{I} \right)^{-1} \tag{18}$$

where $\beta$ is a regularization coefficient, $I$ is the identity matrix, and $X$ is the concatenated hidden states of all samples.

In BrainPy, the reservoir model trained with ridge regression can be implemented as:

```
1 model =ESN(num_in, num_rec, num_out)
2 trainer =bp.RidgeTrainer(model)
3 trainer.fit([X, Y])
4 outputs =trainer.predict(X)
```

## Training with FORCE learning

Ridge regression is an offline learning method, meaning that it needs all of the data to be present before it can learn the model parameters. This can be a problem when training reservoir models with a lot of data, because the memory requirements can be too high for some devices. FORCE learning (*Sussillo and Abbott, 2009*), on the other hand, is an online learning algorithm. This means that it can learn the model parameters one sample of data at a time. This makes it a much more efficient way to train reservoir models with large datasets. Therefore, FORCE learning is a good choice for training reservoir models when the amount of data is large or the memory resources are limited.

The FORCE learning is done using the recursive least squares (RLS) algorithm. It is a supervised error-driven learning rule, that is the weight changes depending on the error made by each neuron: the difference between the output of a neuron $\mathbf{y}_i(t)$ and a desired value $\mathbf{y}_i^{target}(t)$.

$$e_i(t) = \mathbf{y}_i(t) - \mathbf{y}_i^{target}(t) \tag{19}$$

Contrary to the delta learning rule which modifies weights proportionally to the error and to the direct input to a synapse ($\Delta w_{ij} = -\eta \cdot e_i \cdot \mathbf{x}_j$), the RLS learning uses a running estimate of the inverse correlation matrix of the inputs to each neuron:

$$\Delta \mathbf{W}_{out}^{ij} = -e_i \sum_k P_{jk}^i \cdot \mathbf{x}_k \tag{20}$$

Each neuron $i$ therefore stores a square matrix $P^i$, whose size depends on the number of weights arriving at the neuron. Readout neurons receive synapses from all $N$ recurrent neurons, so the $P$ matrix is $N * N$.

The inverse correlation matrix $P$ is updated at each time step with the following rule:

$$\Delta P_{jk}^i = -\frac{\sum_m \sum_n P_{jm}^i \cdot \mathbf{x}_m \cdot \mathbf{x}_n \cdot P_{nk}^i}{1 + \sum_m \sum_n \mathbf{x}_m \cdot P_{mn}^i \cdot \mathbf{x}_n} \tag{21}$$

Each matrix $P^i$ is initialized to the diagonal matrix and scaled by a factor $1/\delta$, where $\delta$ is 1 in the current implementation and can be used to modify implicitly the learning rate (*Sussillo and Abbott, 2009*).

In BrainPy, the reservoir model trained with FORCE learning can be implemented as:

```
1 model =ESN(num_in, num_rec, num_out)
2 trainer =bp.ForceTrainer(model, alpha =0.1)
3 trainer.fit([X, Y])
4 trainer.fit([X, Y])
5 outputs =trainer.predict(X)
```

## Training with backpropagation algorithm

The readout layer in the reservoir model is just a single-layer Perceptron. To train its weights, we can use the backpropagation algorithm. Backpropagation is a method used in artificial neural networks to calculate a gradient that is needed in the calculation of the weights to be used in the network. The loss function used here is the mean square error between the reservoir output and the target output:

$$E = \frac{1}{2} \sum_j (\mathbf{y}_j - \mathbf{y}_j^{\text{target}})^2 \tag{22}$$

The updated weight between neuron $i$ and $j$ is calculated by:

$$\Delta \mathbf{W}_{out}^{ij} = \frac{\partial E}{\partial \mathbf{y}_i} \frac{\partial \mathbf{y}_i}{\partial \mathbf{x}_j}. \tag{23}$$

In BrainPy, the reservoir model trained with backpropagation algorithms can be implemented as:

```
1 reservoir = bp.dyn.Reservoir(num_in, num_rec)
2 readout = bp.dnn.Dense(num_rec, num_out, mode =bm.training_mode)
3 trainer = bp.BPFF(target =readout,
4                   loss_fun =bp.losses.mean_squared_error,
5                   optimizer =bp.optim.Adam(1e-3))
6 # batch_data: the data generated by "reservoir"
7 trainer.fit(batch_data, num_epoch =100)
```

## Evaluation metrics

The performance of a reservoir computing model is usually measured with the mean squared error, that is, the average squared difference between the predicted and actual values:

$$E\left(\mathbf{y}, \mathbf{y}^{\text{target}}\right) = \frac{1}{N_{\text{y}}} \sum_{i=1}^{N_{\text{y}}} \sum_{n=1}^{T} \left(y_i(n) - y_i^{\text{target}}(n)\right)^2, \tag{24}$$

where $N_y$ is the number of training samples and $T$ is the number of time steps.

## Appendix 11

### Experimental details for benchmark comparisons

In this section, we provide a comprehensive overview of the experimental benchmark details used in *Figure 6*, *Figure 7*, and *Figure 8*. The purpose is to present a complete picture of the experimental setup and methodology employed in our study. The details encompass the following aspects: hardware specifications, software versions, algorithm hyperparameters, and performance measurements.

By providing these experimental benchmark details, we aim to ensure transparency and reproducibility, allowing readers and researchers to understand and replicate our experiments accurately.

### Hardware specifications

We conducted a series of experiments on various computing devices, namely the CPU, GPU, and TPU, in order to validate and compare the simulation speeds of several widely utilized brain simulators. The brain simulators under investigation included Brian2 (*Stimberg et al., 2019*), NEURON (*Hines and Carnevale, 1997*), NEST (*Gewaltig and Diesmann, 2007*), GeNN (*Yavuz et al., 2016*), and Brian2CUDA (*Alevi et al., 2022*).

Particularly, the computing devices we used here are:

- CPU: The CPU used was the Intel Xeon W-2255 processor. This is a 10-core, 20-thread CPU based on Intel's Cascade Lake microarchitecture. It has a base clock frequency of 3.7 GHz and a max turbo boost up to 4.5 GHz. The Xeon W-2255 utilizes the LGA2066 socket and supports up to 512 GB of ECC-registered DDR4-2933 RAM across 6 channels. It has 24.75 MB of L3 cache. As a server-grade CPU with a high core count, the Xeon W-2255 is well suited for heavily parallelized simulations.
- GPU: The GPU used was an NVIDIA RTX A6000. This is a high-end Ampere architecture GPU aimed at professional visualization, AI, and compute workloads. The RTX A6000 has 10,752 CUDA cores, 336 tensor cores, and 84 RT cores. It comes equipped with 48 GB of GDDR6 memory clocked at 16 Gbps, providing up to 1 TB/s of memory bandwidth. The RTX A6000 has a maximum power consumption of 300 W and requires auxiliary power connectors. It uses a PCIe 4.0x16 interface. With its large number of CUDA cores and abundant memory, the RTX A6000 is well suited for massively parallel simulations and neural network training.
- TPU: The TPU simulations leveraged the Kaggle free TPU v3-8 cloud instance. This provides access to one of Google's TPU v3 chips via their cloud platform. Specifically, the v3-8 instance gives 8 TPU v3 cores, each providing 128 GB/s of bandwidth to high-performance HBM memory. The TPU v3 is optimized for both training and inferencing of DNNs, providing up to 420 TFLOPS of mixed precision computing. By utilizing Google's cloud TPUs, researchers can run models with minimal coding changes while leveraging Google's optimized deep learning frameworks.

### Software versions

We carried out benchmark experiments using Python 3.10.12 running on Ubuntu 20.04.6 LTS. For any experiments utilizing a GPU, we used version 11.6 of the NVIDIA CUDA Toolkit to take advantage of the parallel processing capabilities of the GPU hardware. Other dependent software versions are: NumPy 1.24.3, Numba 0.57.1, jax 0.4.16, and jaxlib 0.4.16.

The comparison brain simulators for benchmarking are:

- NEURON at version 8.2.0
- NEST at version 3.6
- Brian2 at version 2.5.4
- GeNN at version 4.8.1
- Brian2GeNN at version of 1.7.0: Brian2GeNN is a bridge between Brian2 and GeNN that allows Brian2 users to run their simulations on GPUs using GeNN.
- Brian2CUDA at version of 1.0a3: Brian2CUDA is a native CUDA implementation of Brian2 that allows Brian2 users to run their simulations on GPUs without the need for a bridge.

## Performance measurements

By testing against these simulators on benchmark tasks, we aimed to thoroughly evaluate the performance in terms of *simulation speed* and *accuracy* across different model sizes and paradigms.

When evaluating simulation speed, we focused solely on measuring the execution time of the simulation itself, excluding any time spent on compilation, initialization, or other preparatory steps. This allowed us to make a direct comparison of the raw simulation performance between our simulator and others. We simulated networks of varying sizes. This range of model scales allowed us to determine how well our simulator performs as network size and complexity increase. The final experimental results can be obtained in *Figure 6*, *Figure 7*, and *Figure 8*.

## Accuracy evaluations

To evaluate the accuracy, we configured all simulators to use identical model parameters for a fair comparison.

First, we verified that all simulators generated consistent average firing rates for a given network model (see *Appendix 11—figure 1* and *Appendix 11—figure 2*).

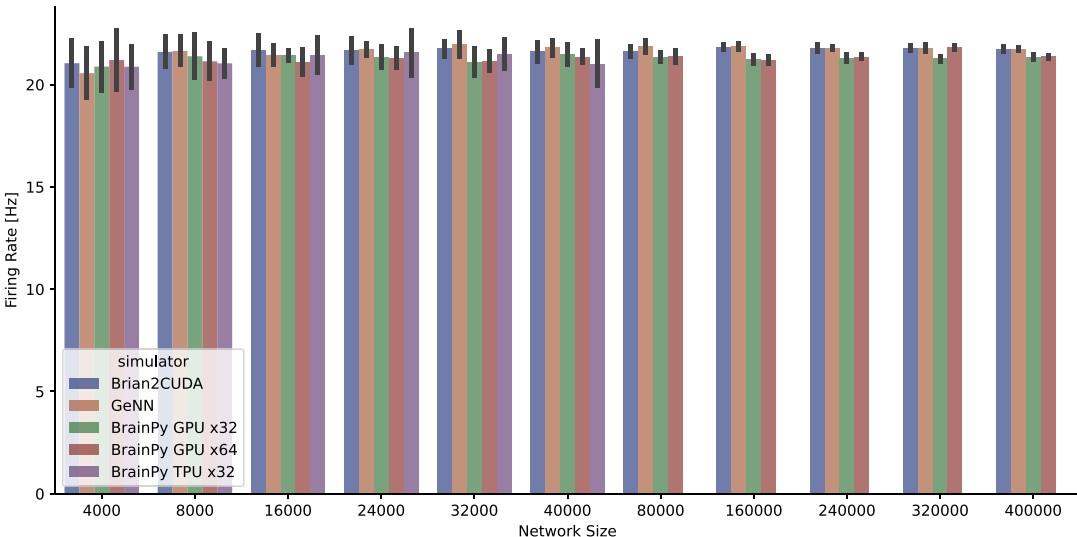

**Appendix 11—figure 1.** The average firing rate per neuron of the COBA network model (*Vogels and Abbott, 2005*) was measured across various simulators running on both GPU and TPU devices. However, it should be noted that the BrainPy TPU x32 simulation was limited to 40,000 neurons due to memory constraints, resulting in a truncated network size.

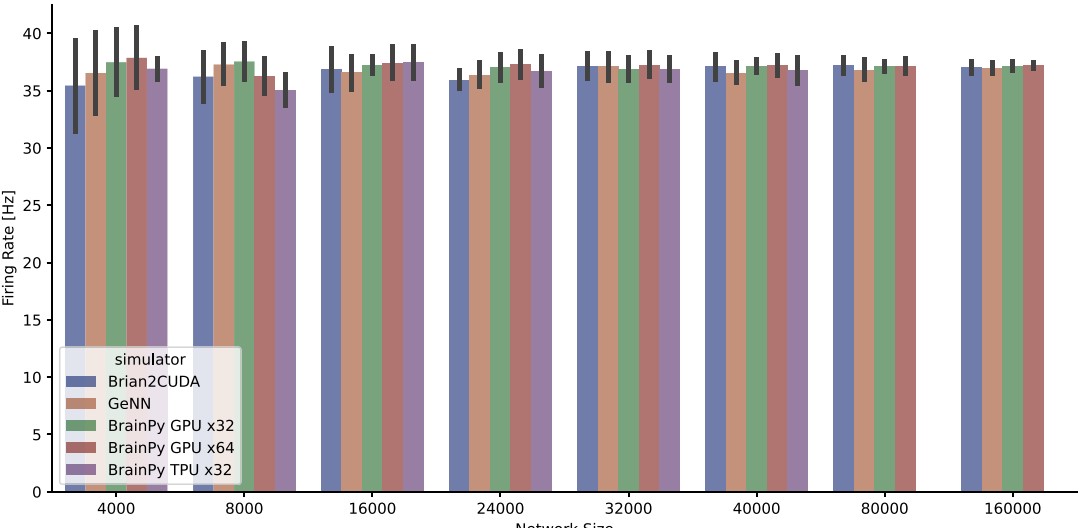

**Appendix 11—figure 2.** The average firing rate per neuron of the COBAHH network model (*Brette et al., 2007*) was measured across various simulators running on both GPU and TPU devices. However, it should be noted that the BrainPy TPU x32 simulation was limited to 40,000 neurons due to memory constraints, resulting in a truncated network size.

Second, we qualitatively compared the overall network activity patterns produced by each simulator to ensure they exhibited the same dynamics. While exact spike-to-spike reproducibility was not guaranteed between different simulator implementations, we confirmed that our simulator produced activity consistent with the reference simulators for both firing rates and network-level dynamics (see *Appendix 11—figure 3* and *Appendix 11—figure 4*).

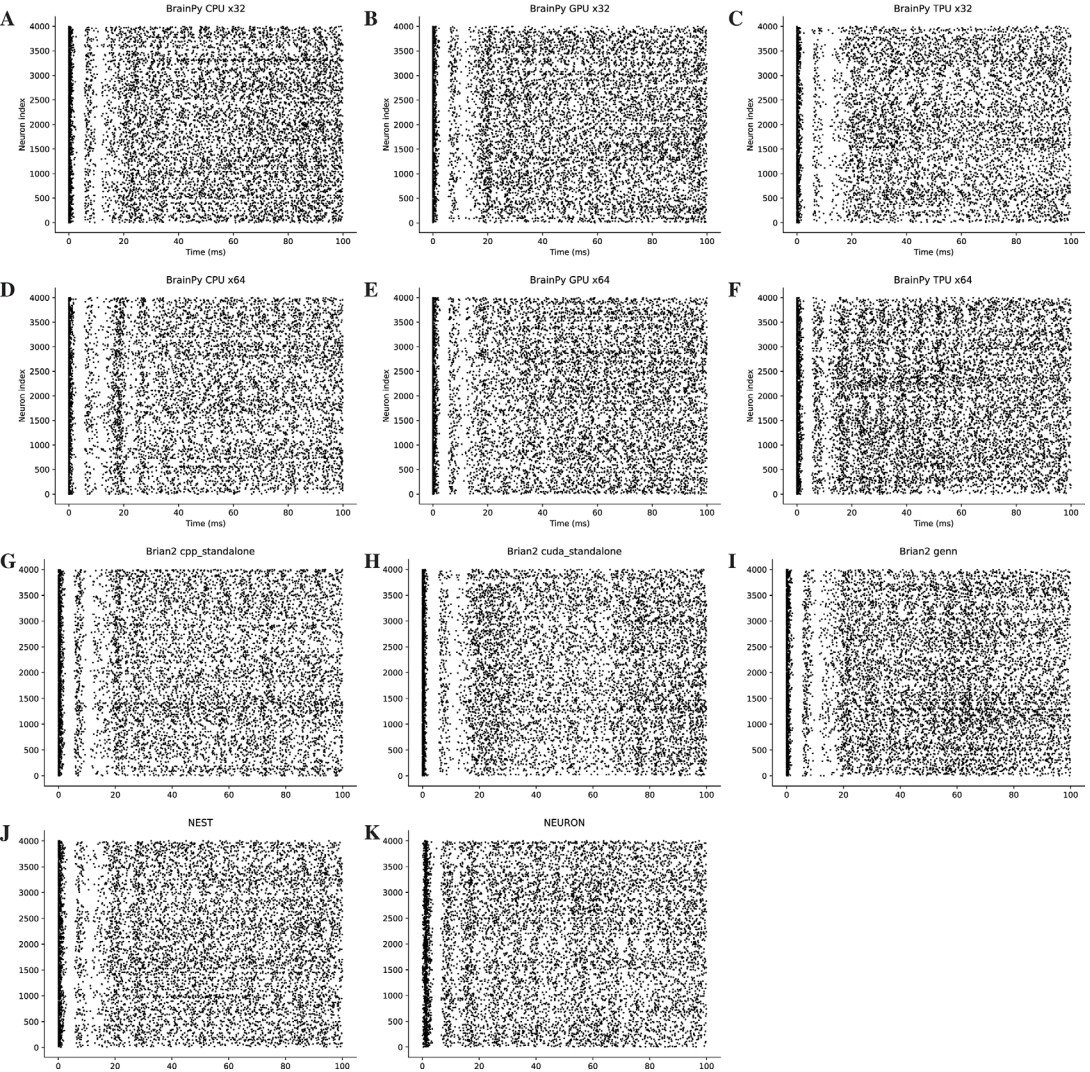

**Appendix 11—figure 3.** Rater diagrams of COBA network model with 4000 neurons (*Vogels and Abbott, 2005*) across different simulators on CPU, GPU, and TPU devices. Here, we only present the simulation results for the initial 100 ms of the experiment.

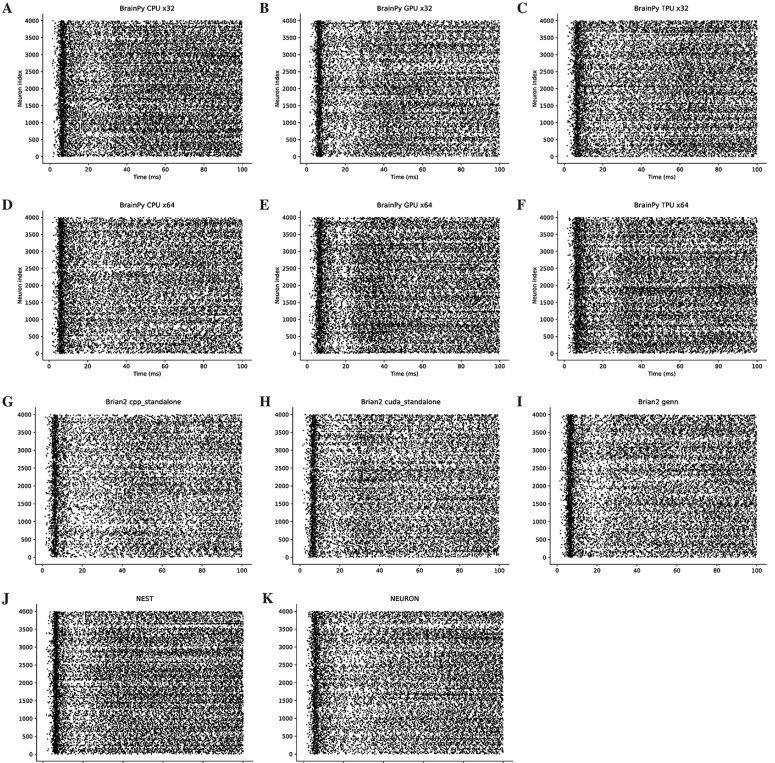

**Appendix 11—figure 4.** Rater diagrams of COBAHH network model with 4000 neurons (*Brette et al., 2007*) across different simulators on CPU, GPU, and TPU devices. Here, we only present the simulation results for the initial 100 ms of the experiment.

Third, we recognize that the precision of numerical computation plays a crucial role in accurately simulating biologically detailed neural networks, such as the COBAHH network model used in our benchmarks. To assess the numerical integration accuracy of each neuron in the COBAHH network model, we conducted a 5-s simulation and examined the presence of NaN membrane potentials at the end of the simulation. The occurrence of NaN membrane potentials indicates that the corresponding neurons are no longer active and signifies a loss of simulation accuracy. This analysis was performed on the GPU backend of Brian2GeNN, Brian2CUDA, and BrainPy. The results of this analysis can be found in *Appendix 11—figure 5*. We specifically focused on the analysis results of single-precision floating-point simulations, as we did not encounter any NaN results when simulating with double precision. Our observations reveal that the single-precision computation with XLA in BrainPy exhibits a higher proportion of NaN results compared to Brian2CUDA and GeNN. Notably, GeNN demonstrated very few occurrences of NaN membrane potentials after simulation, which may be attributed to the special handling of NaN within the GeNN computation.

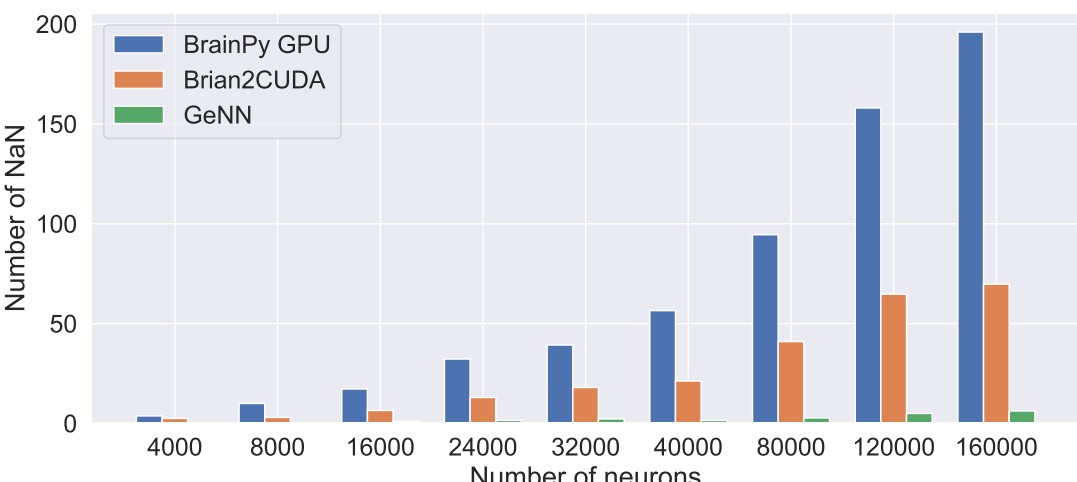

**Appendix 11—figure 5.** Number of neurons with NaN membrane potential after performing a 5-s long simulation of the COBAHH network model using the single-precision floating point.

To address the issue of a larger number of NaN membrane potentials leading to reduced neuronal spiking and communication spikes, we took steps to resolve this problem in BrainPy. Our aim was to ensure a fair comparison when benchmarking the simulation speed against other brain simulators. Specifically, the occurrence of NaN membrane potentials was attributed to the use of the $x/(\exp(x) - 1)$ operation during the integration of the Hodgkin–Huxley neuron model with low-precision floating-point calculations. In order to mitigate the catastrophic loss of precision when $x$ is close to zero, we replaced this operation with the relative error exponential function, represented by $1/\mathrm{exprel}(x)$. This modification ensures that the numerical calculations do not result in NaN values even during significantly long simulations. As a result of this fix, we did not encounter any instances of NaN membrane potentials, even after conducting extensive and prolonged simulations. We believe that these revised operations contribute to a more equitable benchmarking process, eliminating any potential bias caused by the presence of NaN values. The speed benchmarking can be obtained in *Figure 8*.

## Experimental settings
In the below, we describe the experimental setting details used in *Figure 6*, *Figure 7*, and *Figure 8*.

### Figure 6
In the experiments depicted in *Figure 6A*, we exercise precise control over the equivalent theoretical FLOPs (floating-point operations) performed by the LIF (Leaky Integrate-and-Fire) neurons and the matrix–vector multiplication $\mathbf{M}\mathbf{v}$ ($\mathbf{W} \in \mathbb{R}^{m \times m}$ and $\mathbf{v} \in \mathbb{R}^{m}$). In each trial, subsequent to determining the size $m$, we modify the number of neurons in the LIF simulation, ensuring that they collectively execute the same theoretical FLOPs. To simplify the computation of the total FLOPS, we adopt the Euler method with a single time step to solve the differential equations within the LIF neuron model.

On the other hand, for the COBA network (*Vogels and Abbott, 2005*) experiments showcased in *Figure 6B* and *Figure 6C*, the dynamical equations were resolved using the Exponential Euler method with a step size of 0.1ms. The synaptic computation was performed through dense matrix multiplication. Given the presynaptic spikes represented by the vector $\mathbf{v}$, the postsynaptic conductance ($\mathbf{g}_{post}$) is computed using the equation $\mathbf{g}_{post} = g_{max} * \mathbf{v} \cdot M$, where $g_{max}$ denotes the maximum synaptic conductance, and $M$ represents the connection matrix. We assess the simulation time of the network using the aforementioned dense operation under two conditions: with JIT compilation and without JIT compilation.

.

## Figure 7

The COBA network model (***Vogels and Abbott, 2005***) is simulated using the Exponential Euler numerical integration method to approximate solutions to the differential equations governing network dynamics. A fixed simulation time step of 0.1 ms is utilized for numerical simulations.

Accurately isolating the computational time spent on simulating synaptic connections in the COBA network is challenging, as the synapses are computed in an event-driven manner based on spiking activity. Since the number of spikes generated in the network varies across simulations, this causes variability in synapse simulation time measurements. To separately quantify time spent on neuron versus synapse computations, we first profile the neuron simulation time in isolation. This is measured by simulating only the neuronal dynamics without any synaptic connections for a 500-ms duration. Next, we profile the end-to-end run time of the full COBA network simulation including both neurons and dynamic synapses for the same 500-ms duration. Finally, the computational time attributed specifically to simulating synapses can be directly estimated by subtracting the isolated neuron simulation time from the full network simulation time. This approach separates and quantifies the contributions of simulating neuronal and synaptic computations to the overall run time of COBA network simulations.

For the model `without dedicated OP`, we employ dense matrix multiplication to perform the synaptic computation, same as the operation used in ***Figure 6***. On the other hand, For the model `with dedicated OP`, we utilize our event-driven operator called `brainpy.math.event.csrmv` for handling the synaptic computation. This specialized operator is designed specifically to efficiently handle such computations in an event-driven manner.

All simulations were performed using a single thread, excluding the acceleration effect of multi-threading concurrence.

## Figure 8

Same as ***Figure 7***, the COBA network model (***Vogels and Abbott, 2005***), the COBAHH network model (***Brette et al., 2007***), and the decision-making network model (***Wang, 2002***) are simulated using the Exponential Euler numerical integration method with a fixed simulation time step of 0.1ms.

The simulations were conducted across a diverse range of computing devices (including CPUs, GPUs, and TPUs), encompassing various hardware configurations, and were executed using networks of different sizes. To ensure statistical robustness, each trial was repeated 10 times, thereby totaling 10 simulations for each experimental setup. These simulations were carried out for a duration of 5 s. Finally, to provide a representative measure, we reported the average time taken across these 10 simulations.

On the CPU device, we simulate the model using a single thread across different simulators. On Brian2, we open 12 threads for parallel simulation. However, we did not report the simulation speed results of Brian2 runs on the COBA network model, since single-threaded Brian2 runs were faster than using multi-threaded compartments in this case.

## Other supplements

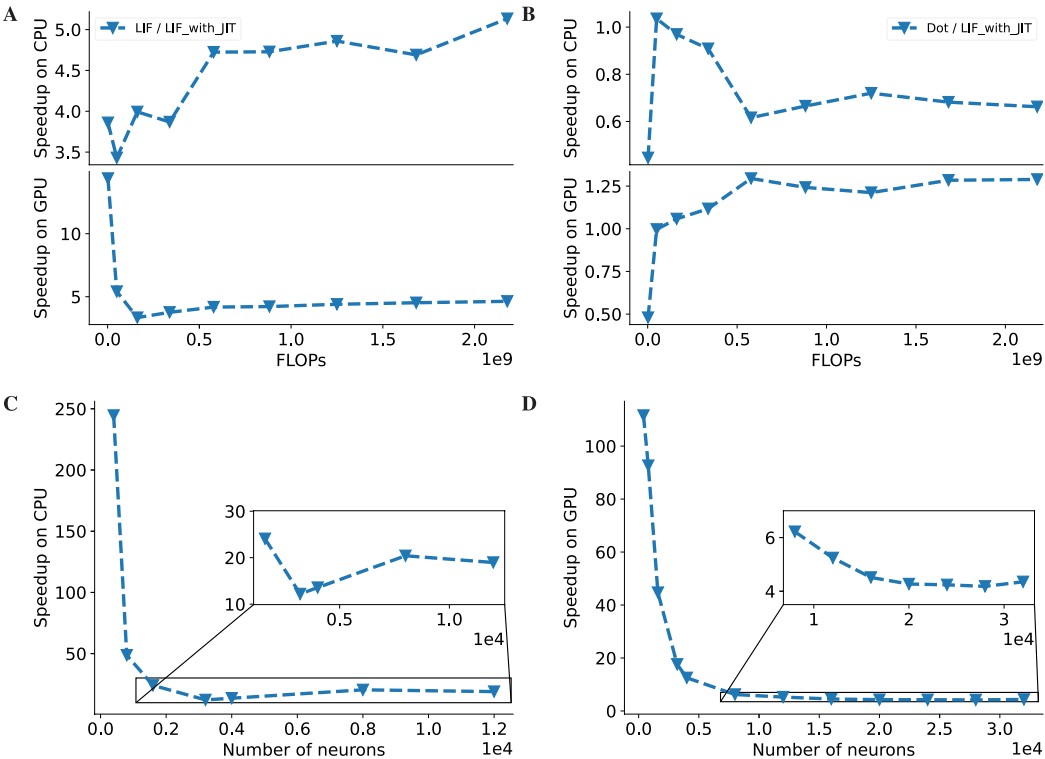

**Appendix 11—figure 6.** The acceleration ratio of JIT compilation on LIF neuron models (***Abbott, 1999***) and COBA network models (***Vogels and Abbott, 2005***). (**A**) The acceleration ratio of just-in-time (JIT) compilation on the LIF neuron model is shown in the plot. The plotted values represent the simulation time ratios of the LIF neuron without JIT and with JIT. The top panel illustrates the acceleration on the CPU device, while the bottom panel demonstrates the acceleration on the GPU device. The acceleration ratios on both devices are approximately five times faster. (**B**) The simulation time ratio of the dense operator compared to the LIF neuron model with JIT compilation is shown. The top panel displays the time ratio on the CPU device, and the bottom panel demonstrates the time ratio on the GPU device. The simulation time ratios on both devices are nearly one, indicating that the jitted LIF neuron, with the same number of floating-point operations (FLOPs) as the 'Dot' operation, can run equivalently fast. (**C**) The acceleration ratio of the COBA network model (***Vogels and Abbott, 2005***) with JIT compilation compared to the model without JIT compilation on the CPU device is shown. The plot demonstrates a tenfold increase in speed after JIT compilation on the CPU device. (**D**) Similar to (**C**), the experiments were conducted on the GPU. Please refer to ***Figure 6*** for the original data.

We evaluate the speedup of event-driven operators over traditional dense matrix operators on both CPU and GPU devices. Evaluation results are listed in ***Appendix 11—figure 7***. Both CPU and GPU devices demonstrate a significant speed advantage of several orders of magnitude when utilizing event-driven operators, in comparison to dense ones. Notably, as the number of neurons increases, event-driven operators showcase an even greater speedup on both platforms.

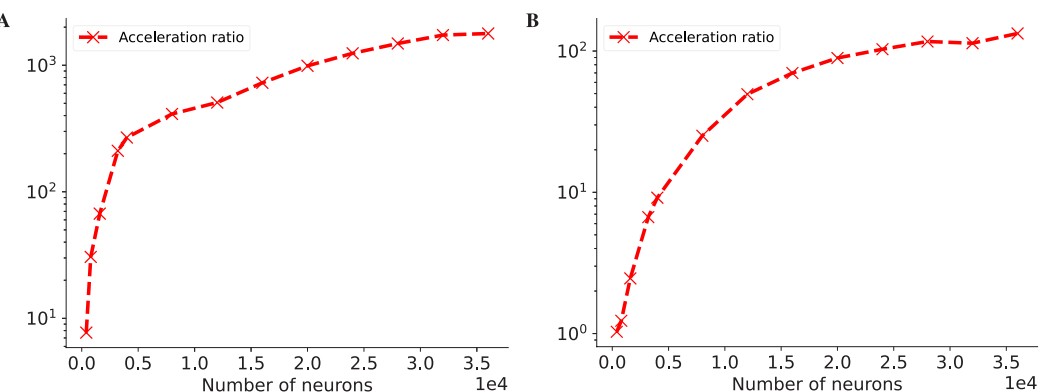

**Appendix 11—figure 7.** The acceleration ratio of dedicated operators of COBA network model (*Vogels and Abbott, 2005*) on both CPU (**A**) and GPU (**B**) devices. Please refer to *Figure 7* for the original data.

We also conducted COBA and COBAHH experiments on TPU devices. The experimental results are shown in *Appendix 11—figure 8*. Although TPUs can perform double precision operations, they are not as efficient at it as they are at lower precisions such as float16 or bfloat16. Therefore, we here only tested models with single-precision operations.

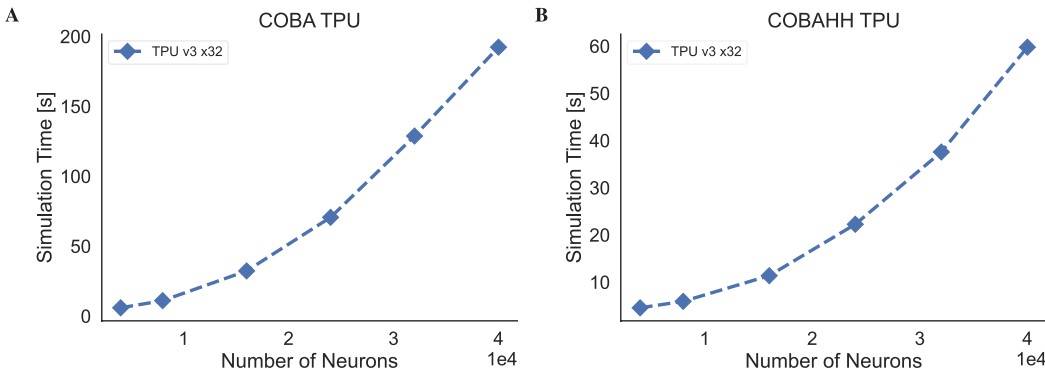

**Appendix 11—figure 8.** The simulation speeds of BrainPy on kaggle TPU v3-8 device. (**A**) The simulation time of the COBA (*Vogels and Abbott, 2005*) network model across different network sizes. (**B**) The simulation time of the COBAHH (*Brette et al., 2007*) network model across different network sizes.

