## [Editor Report]

The paper introduces a new, important framework for neural modelling that promises to offer efficient simulation and analysis tools for a wide range of biologically-realistic neural networks. It provides convincing support for the ease of use, flexibility, and performance of the framework, and features a solid comparison to existing solutions in terms of accuracy. The work is of potential interest to a wide range of computational neuroscientists and researchers working on biologically inspired machine learning applications.

---

## [Decision Letter]

**Decision letter after peer review:**

Thank you for submitting your article "BrainPy: a flexible, integrative, efficient, and extensible framework for general-purpose brain dynamics programming" for consideration by *eLife*. Your article has been reviewed by 3 peer reviewers, one of whom is a member of our Board of Reviewing Editors, and the evaluation has been overseen by Panayiota Poirazi as the Senior Editor.

Essential revisions:

1) Improve the benchmarks and examples. The examples demonstrating the versatility of the simulator lack details (e.g. number of neurons, training method where applicable, etc.). Similarly, the benchmarking procedures are not sufficiently described, and it is not clear whether the results across simulators are comparable. In particular: What are the integration methods, simulation time steps, and floating point precision that were used, and are they identical across simulators? What time is measured exactly (total time including model construction time, synapse creation time, compilation time, or only simulation run time)? What acceleration modes offered by other simulators (multi-threading, MPI, etc.) were used? Were all efforts taken to optimize the code used for other simulators, or is it based on code provided by the developers of those tools? Finally, please scale networks in a way that keeps network activity roughly constant.

2) Verify/demonstrate the accuracy of the simulator. Please provide evidence that the chosen approach does not trade improved performance against reduced accuracy – or if it does, discuss the trade-off.

3) Improve the discussion of limitations and weaknesses of the presented tool, and beware "persuasive communication devices" (https://elifesciences.org/articles/88654). Some of the comparisons to existing state-of-the-art tools do not do justice to their capabilities/advantages.

4) Clarify the target audience of the tool. Is BrainPy primarily intended for computational neuroscientists or machine learning research, or is it particularly aimed at neuro-AI practitioners bridging the two domains? For "pure machine learning", would the authors consider their tool giving an advantage over using a dedicated tool like PyTorch? For computational neuroscience research, the tool does seem to lack essential local plasticity rules (e.g. Hebbian plasticity or STDP – the latter is mentioned in Appendix 8 but refers to a non-existing class in the package).

*Reviewer #1 (Recommendations for the authors):*

General remarks:

– The package contains automatic unit/integration tests, but from a cursory look, they (many? most?) only test for the absence of basic errors (i.e. the code does not fail to run), not for the correctness of the result. Some classes (e.g. GapJunction) never seem to be tested at all.

– Personally, I'd remove the supplementary document, it contains information that could rather go into the documentation, since it risks being outdated quickly.

Specific remarks:

– In the abstract, the sentence "Brain dynamics modeling is an indispensable tool for […] modeling the dynamics of the neural circuits" sounds rather redundant/tautological.

– l.36: "An example …" introduces two examples (TensorFlow and PyTorch).

– References to sections seem to be broken (e.g. l. 119, l. 136).

– L.163, "as standalone modules" is not very clear.

– L.177: I'm not sure the authors really mean "recursive" here, shouldn't it be "hierarchical"? Also, l. 181: not sure what the authors exactly mean here, in most simulators, you can e.g. simulate a neuron individually but also include it in a network and simulate as part of the network.

– Figure 2A: maybe add to the figure/caption that BrainPy is imported as bp? What does sh refer to?

– Figure 2D: super should refer to System and not Circuit, I think.

– Figure 3E: Typo in title ("Rater" → "Raster").

– L.199: not sure one can say that the figure "demonstrates" that one can simulate models in parallel.

– L.304: this needs to be qualified a bit, I think, since it is also the idea behind code generation used by several other simulators.

– Figure 7A: missing y-axis label.

– Figure 7C: having the ratio in the same plot is an odd choice, I'd at least use a second y-axis to make this clearer.

– Figure 7D: the y-axis label should not mention Ratio (not plotted here).

– L.311/Figure 8C: this (and the results) seems a problem with the code written by the authors for Brian2, maybe contact the Brian2 developers?

– Figure 8C: I'd recommend changing the colors to match the previously used colors to avoid confusion.

– L. 376: in what way can "standard debugging tools" debug issues with JAX/XAL?

– Appendix 1, last paragraph: "Moreover, descriptive languages…" – I do not quite understand this. With descriptive languages, users do not have to use low-level language. Or does this refer to the effort of the tool developers?

– Appendix 8, l. 1141: apart from the "a piece of cake" expression that I'd rather replace, I did not find any STDP class in BrainPy.

*Reviewer #2 (Recommendations for the authors):*

There are a couple of significant points to mention here:

You state on p12: "We also implemented this decision-making model with Brian2 on the CPU (although it was stated that Brian2 could target on GPU via code generation with GeNN (Stimberg et al., 2020), we could not run this model on a GPU with Brian2 successfully) and observed that the running speed of Brian2 was significantly slower than that of BrainPy.". In order to make this a fair test, I would expect more effort in getting the model to run, choose another standard model to run with both platforms, or at least provide a substantial explanation for why you were unable to. The editor is in fact the lead author of Brian2 and would very likely be able to help you get support from the GeNN team.

I was also unable to create a Docker image on an ARM-based Mac since jaxlib is unavailable for the combination of linux on ARM processors. Until this (reasonably common) combination becomes an official target for compiled binaries, it would be good to provide instructions on how to compile it from source or otherwise work around this missing dependency. Similarly, I was unable to install brainpylib.

*Reviewer #3 (Recommendations for the authors):*

More care should be given to the reproducibility of the benchmarks. When using a benchmark like COBA (Figures 6, 7, and 8), the numerical method should be the same, or changes should be stated and explained. The authors should state how many runs they used for each experiment to obtain the presented timings. From the codes in the repository, we assume that they are always analyzing a single run. We would recommend performing simulation multiple times to catch the deviations in-between runs.

The authors use a fixed connection probability of 2% in all scaled benchmarks and to compensate for the increasing number of connections, the authors rescale the weights. However, this approach leads to the following consequence: with an increasing number of connections the activity in the population shrinks. For the settings used in the benchmarks, it would be averaged across 5 runs each: 20.83 +/- 0.87 Hz (4,000 neurons); 10.89 +/- 0.34 Hz (8,000 neurons); 5.25 +/- 0.11 Hz (16,000 neurons); 3.61 +/- 0.04 Hz (24,000 neurons); 2.83 +/- 0.02 Hz (32,000 neurons); 2.38 +/- 0.03 Hz (40,000 neurons).

To retain the average firing rate, one would rescale the probability instead of the weights, e.g., p = 80. / N, where N is the number of neurons while the weight values remain the same in all configurations. Such an approach was used by Stimberg et al. (2020), which has been already cited by the authors. Such an approach retains the firing frequencies at a comparable level, e.g.: 21.91 +/- 1.32 Hz (4,000 neurons); 21.59 +/- 0.71 Hz (8,000 neurons); 21.15 +/- 0.07 Hz (16,000 neurons); 21.33 +/- 0.16 Hz (24,000 neurons); 21.56 +/- 0.19 Hz (32,000 neurons); 21.59 +/- 0.39 Hz (40,000 neurons).

For the COBAHH benchmark, it is first not clear whether the base firing rate of 37 Hz is not too high. Secondly, as the authors used here also the weight rescale, we can see the same drop in activity with increasing network sizes: 37.08 +/- 0.88 Hz (4,000 neurons); 17.57 +/- 0.43 Hz (8,000 neurons); 3.61 +/- 0.01 Hz (16,000 neurons); 3.11 +/- 0.02 Hz (24,000 neurons); 2.83 +/- 0.03 Hz (32,000 neurons); 2.73 +/- 0.05 Hz (40,000 neurons); 2.53 +/- 0.03 Hz (60,000 neurons); 2.55 +/- 0.05 Hz (80,000 neurons); 2.65 +/- 0.04 Hz (120,000 neurons).

[Editors’ note: further revisions were suggested prior to acceptance, as described below.]

Thank you for resubmitting your work entitled "BrainPy, a flexible, integrative, efficient, and extensible framework for general-purpose brain dynamics programming" for further consideration by *eLife*. Your revised article has been evaluated by Panayiota Poirazi (Senior Editor) and a Reviewing Editor.

The manuscript has been improved but there are some remaining issues that need to be addressed, as outlined below:

– The benchmarks, accuracy evaluations and their descriptions have been greatly improved, but they still have shortcomings that at least merit discussion: while the accuracy evaluations in Appendix 11 convincingly argue for no *major* discrepancies between the simulations, they are not enough to unambiguously state that single precision leads to "a significant speedup without the sacrifice of the simulation accuracy". From my personal experience, simulating a large COBAHH network with single precision will inevitably lead to division by 0 for a few neurons, if not using equations that use special functions such as exprel (scipy, brian2), or vtrap (NEURON) to avoid this issue. This will usually not show in the average firing rate or a raster plot, but having neurons that are no longer active with a membrane potential of NaN should certainly be considered a loss of accuracy. I am aware that a more in-depth analysis of the accuracy (as e.g. in van Albada et al. 2018; doi: 10.3389/fnins.2018.00291) is out of scope, but I think a more careful evaluation of the accuracy is warranted. It should also be noted that the raster plots for NEST and NEURON do look quite different from the other results in Appendix 11 figures 3 and 4 – probably due to different initial conditions? Finally, the authors state that "Notably, BrainPy offers support for utilizing single-precision floating-point (x32) arithmetic", but this feature is present in several of the other simulators that were benchmarked as well and has been prominently mentioned in the papers introducing the GPU simulators discussed in the manuscript (GeNN, Brian2GeNN, Brian2CUDA).

– The manuscript (understandably) puts forwards the features of the simulator, but sometimes is not sufficiently making the link to existing literature on the topic and/or implementations in existing simulators. Apart from the single-precision feature mentioned above, this is in particular the case for the sparse and event-driven synaptic connectivity. Implementing such connectivity has been at the heart of simulator development for a long time, in particular in the context of GPU acceleration (for some early examples, see Plesser et al. 2007 http://link.springer.com/chapter/10.1007/978-3-540-74466-5_71; Fidjeland et al. 2009 https://doi.org/10.1109/ASAP.2009.24; Brette and Goodman 2011 https://doi.org/10.1162/NECO_a_00123). Similarly, "just-in-time connectivity" has been implemented before (e.g. Knight and Nowotny 2021 https://doi.org/10.1038/s43588-020-00022-7). Please also note that the review in Appendix 1 should probably mention NMODL for the NEURON simulator (which I would not describe as having "a library of standard models written in C or CUDA").

– The manuscript text is still somewhat hard to follow at times due to jargon that is not common usage (which also relates to the previous point). For example, *eLife* readers are not necessarily familiar with technical terms around just-in-time compilation (e.g. "lowering" code onto CPU/GPU, "fusing" operators), and will therefore have a hard time understanding where the speed benefit of JIT compilation comes from. Terms such as "time delays" and "length delays" (Appendix 4) are also non-standard and need explanation (as a side note: the mechanism for delaying spikes is never explained). Finally, readers might confuse the "event-driven operators" mentioned in the manuscript with event-driven simulation (as opposed to clock-driven; see e.g. Ros et al. 2006 10.1162/neco.2006.18.12.2959). I'd suggest the authors carefully reconsider non-standard terms and clearly define them if no commonly used term exist or if a large part of the readership might not be familiar with them. Other examples of such terms/concepts are "bfloat16" (l. 359), "data parallelism" vs. "model parallelism" (l. 479), or even "TPU" (from the abstract on).

---

## [Author Response]

Essential revisions:1) Improve the benchmarks and examples. The examples demonstrating the versatility of the simulator lack details (e.g. number of neurons, training method where applicable, etc.). Similarly, the benchmarking procedures are not sufficiently described, and it is not clear whether the results across simulators are comparable. In particular: What are the integration methods, simulation time steps, and floating point precision that were used, and are they identical across simulators? What time is measured exactly (total time including model construction time, synapse creation time, compilation time, or only simulation run time)? What acceleration modes offered by other simulators (multi-threading, MPI, etc.) were used? Were all efforts taken to optimize the code used for other simulators, or is it based on code provided by the developers of those tools? Finally, please scale networks in a way that keeps network activity roughly constant.

We have carefully considered these comments and addressed each of these concerns regarding the benchmarks and examples presented in our paper.

1. Lack of Details in Examples: In the revised version of the paper, we provide additional information and any other pertinent details to enhance the clarity and replicability of our results. Particularly, in Appendix 9, we provide the mathematical description, the number of neurons, the connection density, and delay times used in our multi-scale spiking network; in Appendix 10, we provide the detail description of reservoir models, evaluation metrics, training algorithms, and their implementations in BrainPy; in Appendix 11, we elaborate the hardware and software specifications and experimental details for benchmark comparisons.

2. Inadequate Description of Benchmarking Procedures: In the revised paper, particularly, in L328-L329 of the main text at section of "Efficient performance of BrainPy" and in Appendix 11, we elaborate on the integration methods, simulation time steps, and floating-point precision used in our experiments. We also ensure that these parameters are clearly stated and identical across all simulators involved in the benchmarking process, see "Accuracy evaluations" in Appendix 11 (L1550 – L1580).

3. Clarification on Measured Time: In the revised paper, we state that all simulations only measured the model execution time, excluding model construction time, synapse creation time, and compilation time, see "Performance measurements" in Appendix 11 (L1539 – L1548).

4. Consideration of Acceleration Modes: In the revised version, we provide the simulation speed of other brain simulators on different acceleration modes, see Figure 8. For instance, we utilize the fastest possible option --- the C++ standalone mode in Brian2 --- for speed evaluations. Furthermore, we have requested the developers of the comparison simulators for optimizing the benchmark models, ensuring a fair and accurate comparison.

5. Scaling Networks to Maintain Activity: In the revised manuscript, we adopt the suggestion of Reviewer #3 and apply the appropriate scaling techniques to maintain consistent network activity throughout our experiments. These details can be found in Appendix 11 (also see Appendix 11—figure 1 and Appendix 11—figure 2).

2) Verify/demonstrate the accuracy of the simulator. Please provide evidence that the chosen approach does not trade improved performance against reduced accuracy – or if it does, discuss the trade-off.

We agree that an explicit comparison would help alleviate any doubts and provide a more comprehensive understanding of our framework's accuracy. We have revised our manuscript to include a dedicated section, particularly Appendix 11. In this section, we verified that all simulators generated consistent average firing rates for the given benchmark network models (figure 1 and figure 2 in Appendix 11). These verifications were performed under different network sizes (ranging from 4e^3^ to 4e^5^) and different computing platforms (CPU, GPU and TPU). We also qualitatively compared the overall network activity patterns produced by each simulator to ensure they exhibited the same dynamics (figure 3 and figure 4 in Appendix 11). While exact spike-to-spike reproducibility was not guaranteed between different simulator implementations, we confirmed that our simulator produced activity consistent with the reference simulators for both firing rates and network-level dynamics. Additionally, BrainPy did not sacrifice simulation accuracy for speed performance. Despite using single precision floating point, BrainPy was able to produce consistent firing rates and raster diagrams across all simulations (see figure 3 and figure 4 in Appendix 11).

We hope these revisions can ensure that our manuscript provides a clear and robust validation of the accuracy of our simulator.

3) Improve the discussion of limitations and weaknesses of the presented tool, and beware "persuasive communication devices" (https://elifesciences.org/articles/88654). Some of the comparisons to existing state-of-the-art tools do not do justice to their capabilities/advantages.

We agree that our paper could be improved by more clearly stating the limitations of our approach and comparing it to established approaches. We have revised the paper and added two new subsections in the Discussion section to address these specific concerns:

1. The Limitations subsection (L448 – L491) acknowledges restrictions of BrainPy paradigm which uses a Python-based object-oriented programming. It highlights three main categories of limitations: (a) approach limitations, (b) functionality limitations, (c) parallelization limitations. These limitations highlight areas where BrainPy may require further development to improve its functionality, performance, and compatibility with different modeling approaches.

2. The Future Work subsection (L493 – L526) outlines development enhancements we aim to pursue in the near term. It emphasizes the need for further development in order to meet the demands of the field. Three key areas requiring attention are highlighted: (a) multi-compartment neuron models, (b) ultra-large-scale brain simulations, (c) bridging with acceleration computing systems.

In addition to these changes, we have also made a number of other minor changes to the paper to improve its clarity and readability.

4) Clarify the target audience of the tool. Is BrainPy primarily intended for computational neuroscientists or machine learning research, or is it particularly aimed at neuro-AI practitioners bridging the two domains? For "pure machine learning", would the authors consider their tool giving an advantage over using a dedicated tool like PyTorch? For computational neuroscience research, the tool does seem to lack essential local plasticity rules (e.g. Hebbian plasticity or STDP – the latter is mentioned in Appendix 8 but refers to a non-existing class in the package).

We appreciate the reviewer's concern regarding the scope of BrainPy and the need for clarification regarding the target audience.

BrainPy is designed to cater to both computational neuroscientists and neuro-AI practitioners by integrating detailed neural models, classical point models, rate-coded models, and deep learning models. The platform aims to provide a general-purpose programming framework for modeling brain dynamics, allowing users to explore the dynamics of brain or brain-inspired models that combines insights from biology and machine learning.

Particularly, brain dynamics models (provided in brainpy.dyn module) and deep learning models (provided in brainpy.dnn module) are closely integrated with each other in BrainPy. First, to build brain dynamics models, users should use the building blocks in brainpy.dnn module to create synaptic projections. The following code demonstrates how to use brainpy.dnn.Linear to create a synaptic model with dense connections:

class ExponDenseCOBA(bp.Projection): 
    def __init__(self, pre, post, delay, g_max, tau, E):  
        super().__init__()   
        self.proj = bp.dyn.ProjAlignPostMg2(    
            pre=pre,         
            delay=delay,           
            comm=bp.dnn.Linear(pre.num, post.num, g_max), # HERE    
            syn=bp.dyn.Expon.desc(post.num, tau=tau),     
            out=bp.dyn.COBA.desc(E=E),           
            post=post,      
       )

Or, we can use a sparse connection module brainpy.dnn.EventCSRLinear to create a synaptic model with sparse connections and event-driven computations.

class ExponDenseCOBA(bp.Projection):  
    def __init__(self, pre, post, delay, prob, g_max, tau, E): 
        super().__init__()    
        self.proj = bp.dyn.ProjAlignPostMg2(      
             pre=pre,      
             delay=delay,      
             comm=bp.dnn.EventCSRLinear(bp.conn.FixedProb(prob, pre=pre.num, post=post.num), g_max), # HERE
               syn=bp.dyn.Expon.desc(post.num, tau=tau),      
             out=bp.dyn.COBA.desc(E=E),      
             post=post,    
        )

Or, we can use the convolutional neural network to create a synaptic model with convolutional connections.

class ExponDenseCOBA(bp.Projection): 
    def __init__(self, pre, post, delay, prob, g_max, tau, E):  
        super().__init__()  
        self.proj = bp.dyn.ProjAlignPostMg2(     
            pre=pre,     
            delay=delay,     
            # HERE  
             comm=bp.Sequntial(         
                    lambda a: a.reshape(pre.size), # reshape to a 4D array      
                 bp.dnn.Conv2D(1, 10, 3), # convolution operator     
                    lambda a: a.flatten(), # flatten to a 1D vector      
             ),  
             syn=bp.dyn.Expon.desc(post.num, tau=tau),  
             out=bp.dyn.COBA.desc(E=E),  
             post=post,    
        )

Second, to build brain-inspired computing models for machine learning, users could also take advantages of neuronal and synaptic dynamics have been provided in brainpy.dyn module.

To that end, BrainPy provides building blocks of detailed conductance-based models like Hodgkin-Huxley, as well as common deep learning layers like convolutions.

Regarding the advantage of using BrainPy over PyTorch for purely deep networks, we acknowledge that existing deep learning libraries like Flax in the JAX ecosystem provide extensive tools and examples for constructing traditional deep neural networks. While BrainPy does implement standard deep learning layers, our primary focus is not to compete directly with those libraries. Instead, we provide these models for the seamless integration of deep learning layers within BrainPy's core modeling abstractions, including variables and dynamical systems. This integration allows researchers to incorporate common deep learning layers into their brain models. Additionally, the inclusion of deep learning layers in BrainPy serves as examples for customization and facilitates the development of tailored layers for neuroscience research. Researchers can modify or extend the implementations to suit their specific needs.

In summary, BrainPy's scope focuses on the general-purpose brain dynamics programming. The target audience includes computational neuroscientists who want to incorporate insights from machine learning, as well as some ML researchers interested in integrating brain-like components.

Reviewer #1 (Recommendations for the authors):General remarks:– The package contains automatic unit/integration tests, but from a cursory look, they (many? most?) only test for the absence of basic errors (i.e. the code does not fail to run), not for the correctness of the result. Some classes (e.g. GapJunction) never seem to be tested at all.

We apologize for any inconvenience caused by the limited testing scope initially. Previously, our main focus for testing was on basic elements within the infrastructure modules, such as brainpy.math and brainpy.integrators. We appreciate the reviewer bringing this limitation to our attention. We have devoted to improving the testing coverage to ensure the reliability and accuracy of the package. As part of these improvements, we have added new test suites for models in brainpy.rates, brainpy.neurons, brainpy.synapses, brainpy.dnn, brainpy.dyn, and brainpy.channels modules.

Reviewer #2 (Recommendations for the authors):There are a couple of significant points to mention here:You state on p12: "We also implemented this decision-making model with Brian2 on the CPU (although it was stated that Brian2 could target on GPU via code generation with GeNN (Stimberg et al., 2020), we could not run this model on a GPU with Brian2 successfully), and observed that the running speed of Brian2 was significantly slower than that of BrainPy.". In order to make this a fair test, I would expect more effort in getting the model to run, choose another standard model to run with both platforms, or at least provide a substantial explanation for why you were unable to. The editor is in fact the lead author of Brian2 and would very likely be able to help you get support from the GeNN team.I was also unable to create a Docker image on an ARM-based Mac since jaxlib is unavailable for the combination of linux on ARM processors. Until this (reasonably common) combination becomes an official target for compiled binaries, it would be good to provide instructions on how to compile it from source or otherwise work around this missing dependency. Similarly, I was unable to install brainpylib.

We appreciate the reviewer's feedback and acknowledge the issues they encountered during their evaluation. We have made the necessary revisions to address the concerns raised.

First, the Docker image has been provided. Users can directly install brainpy through docker pull brainpy.

Second, we have given a detailed installation instruction on ARM-based Mac in our official website (https://brainpy.readthedocs.io/en/latest/quickstart/installation.html#). The most important thing is that users should first install the latest M1 macOS version of Miniconda or Anaconda.

Third, brainpylib can be directly installed on ARM-based Mac through pip install brainpylib if the M1 macOS version of conda is installed.

Reviewer #3 (Recommendations for the authors):More care should be given to the reproducibility of the benchmarks. When using a benchmark like COBA (Figures 6, 7, and 8), the numerical method should be the same, or changes should be stated and explained. The authors should state how many runs they used for each experiment to obtain the presented timings. From the codes in the repository, we assume that they are always analyzing a single run. We would recommend performing simulation multiple times to catch the deviations in-between runs.

To address this concern, we add a new appendix to explicitly state all experiments in this paper (please refer to Appendix 11).

Additionally, we apologize for the lack of information regarding the number of runs performed for each experiment. We have performed 10 runs of each benchmark and updated the paper with the average execution timings (see L1634 – L1637). Please find the updated figures reflecting these changes in our revised paper (Figure 8).

[Editors’ note: further revisions were suggested prior to acceptance, as described below.]

The manuscript has been improved but there are some remaining issues that need to be addressed, as outlined below:– The benchmarks, accuracy evaluations and their descriptions have been greatly improved, but they still have shortcomings that at least merit discussion: while the accuracy evaluations in Appendix 11 convincingly argue for no major discrepancies between the simulations, they are not enough to unambiguously state that single precision leads to "a significant speedup without the sacrifice of the simulation accuracy". From my personal experience, simulating a large COBAHH network with single precision will inevitably lead to division by 0 for a few neurons, if not using equations that use special functions such as exprel (scipy, brian2), or vtrap (NEURON) to avoid this issue. This will usually not show in the average firing rate or a raster plot, but having neurons that are no longer active with a membrane potential of NaN should certainly be considered a loss of accuracy. I am aware that a more in-depth analysis of the accuracy (as e.g. in van Albada et al. 2018; doi: 10.3389/fnins.2018.00291) is out of scope, but I think a more careful evaluation of the accuracy is warranted.

We appreciate the reviewer for bringing up the precision issue once again. We have carefully considered this feedback and taken several steps to address this concern.

Firstly, we conducted additional analysis to evaluate the x32 computation accuracy of BrainPy, Brian2CUDA, and GeNN. This analysis, depicted in Appendix 11—figure 5, focused on examining the number of neurons displaying NaN membrane potentials after a 5-second COBAHH simulation. Interestingly, we observed that BrainPy exhibited a higher number of NaN neurons compared to the other frameworks (see L1670-L1684).

To rectify this issue and ensure a fair speed benchmark, we replaced the problematic operation of x/(exp⁡(x)−1) with 1/exprel(x) when integrating the Hodgkin-Huxley neuron models using single-precision floating-point. By making this adjustment, we successfully eliminated the occurrence of NaN neurons after the COBAHH simulation (see L1689-L1702). We have also updated the new speed data for BrainPy x32 computation in Figures 8B and 8E to reflect these changes.

Furthermore, we conducted an extensive analysis of the speed performance of Brian2, Brian2CUDA, and GeNN, specifically focusing on the x32 computation. Similar to the BrainPy x32 experiments, we performed these analyses on the three benchmark scenarios. The updated speed data for Brian2, Brian2CUDA, and GeNN with the x32 computation are included in Figure 8 to provide a more comprehensive comparison.

We hope that these efforts have improved the quality and accuracy of our research. We greatly appreciate the reviewer for drawing our attention to this issue once again, as it has allowed us to enhance the reliability of our results.

It should also be noted that the raster plots for NEST and NEURON do look quite different from the other results in Appendix 11 figures 3 and 4 – probably due to different initial conditions?

We apologize for the discrepancy in the initial conditions when simulating the EI balance network models using NEST and NEURON. We have addressed this issue and updated the reproducing code on our GitHub repository (https://github.com/brainpy/brainpy-paper-reproducibility, see L69 in COBA_pynn.py and L54 in COBAHH_pynn.py). As a result, the corresponding figures, specifically Appendix 11—figure 3/4 J and K, have been revised and updated accordingly.

Finally, the authors state that "Notably, BrainPy offers support for utilizing single-precision floating-point (x32) arithmetic", but this feature is present in several of the other simulators that were benchmarked as well, and has been prominently mentioned in the papers introducing the GPU simulators discussed in the manuscript (GeNN, Brian2GeNN, Brian2CUDA).

We appreciate the reviewer pointing out our misleading statement regarding single-precision floating point (x32) support. To address this point, we have updated the manuscript in the following two ways. Firstly, we conducted additional experiments to benchmark the x32 computation capability of Brain2, Brian2GeNN, and Brian2CUDA. The results of these experiments are now presented in the revised Figure 8. Secondly, we have corrected our statement to highlight that BrainPy's x32 computation offers a higher performance speedup. We hope these changes rectify the inaccuracies and provide a more accurate description of BrainPy's features. Thank you once again for your valuable feedback.

– The manuscript (understandably) puts forwards the features of the simulator, but sometimes is not sufficiently making the link to existing literature on the topic and/or implementations in existing simulators. Apart from the single-precision feature mentioned above, this is in particular the case for the sparse and event-driven synaptic connectivity. Implementing such connectivity has been at the heart of simulator development for a long time, in particular in the context of GPU acceleration (for some early examples, see Plesser et al. 2007 http://link.springer.com/chapter/10.1007/978-3-540-74466-5_71; Fidjeland et al. 2009 https://doi.org/10.1109/ASAP.2009.24; Brette and Goodman 2011 https://doi.org/10.1162/NECO_a_00123). Similarly, "just-in-time connectivity" has been implemented before (e.g. Knight and Nowotny 2021 https://doi.org/10.1038/s43588-020-00022-7). Please also note that the review in Appendix 1 should probably mention NMODL for the NEURON simulator (which I would not describe as having "a library of standard models written in C or CUDA").

We greatly appreciate the reviewer's valuable feedback on improving the literature acknowledgement and accurately reviewing the related works. We sincerely apologize for any carelessness and neglect in our previous statements regarding the acceleration with sparse, event-driven, and just-in-time connectivity. To address this issue, we have made the necessary changes. Specifically, we have thoroughly revised Appendix 4, placing emphasis on acknowledging previous simulator works in this area. We have included citations to notable examples of CPU and GPU sparse/event-driven connectivity support, such as the works of Plesser et al. (2007), Fidjeland et al. (2009), Brette and Goodman (2011), and more. Additionally, we have incorporated citations to Knight and Nowotny (2021) and earlier works on just-in-time connectivity, including Lytton et al. (2008) and Carvalho et al. (2020). Regarding the NEURON review, thank the reviewer for clarifying that NMODL (not C/CUDA) enables model specification. We have corrected this simulator's description accordingly (see Appendix 1).

– The manuscript text is still somewhat hard to follow at times due to jargon that is not common usage (which also relates to the previous point). For example, eLife readers are not necessarily familiar with technical terms around just-in-time compilation (e.g. "lowering" code onto CPU/GPU, "fusing" operators), and will therefore have a hard time understanding where the speed benefit of JIT compilation comes from. Terms such as "time delays" and "length delays" (Appendix 4) are also non-standard and need explanation (as a side note: the mechanism for delaying spikes is never explained). Finally, readers might confuse the "event-driven operators" mentioned in the manuscript with event-driven simulation (as opposed to clock-driven; see e.g. Ros et al. 2006 10.1162/neco.2006.18.12.2959). I'd suggest the authors carefully reconsider non-standard terms and clearly define them if no commonly used term exist or if a large part of the readership might not be familiar with them. Other examples of such terms/concepts are "bfloat16" (l. 359), "data parallelism" vs. "model parallelism" (l. 479), or even "TPU" (from the abstract on).

We thank the reviewer for bringing up the concern regarding the readability for a broader readership. We have taken this into consideration and have made the following revisions to address the jargon issues raised:

1. Just-in-Time Compilation: We have replaced instances of "lowering" with "compiling" or "transforming" throughout the document. Additionally, we have provided an introductory explanation of operator fusion without assuming prior knowledge (L281-L285).

2. Time Delays: In Appendix 5, we have provided clearer definitions for the terms "TimeDelay" and "LengthDelay." We have also included explanations regarding the usage and implementation of delay mechanisms (L1147-L1172, and Appendix 5—figure 1).

3. Event-Driven Operators: We have carefully reviewed the text to separate the concepts of *event-driven operators* and *event-driven simulation*. We have also included an explanation to differentiate between these terminologies. Details please see L1021-L1033.

4. Other Concepts: We now define the concepts of "bfloat16," "data parallelism," "model parallelism," "TPUs," and other relevant terms when they are first introduced in the document.